# High-resolution secretory timeline from vesicle formation at the Golgi to fusion at the plasma membrane in *S. cerevisiae*

**Robert M Gingras\*, Abigail M Sulpizio, Joelle Park, Anthony Bretscher\***

Department of Molecular Biology and Genetics, Weill Institute for Cell and Molecular Biology, Cornell University, Ithaca, United States

**Abstract** Most of the components in the yeast secretory pathway have been studied, yet a high-resolution temporal timeline of their participation is lacking. Here, we define the order of acquisition, lifetime, and release of critical components involved in late secretion from the Golgi to the plasma membrane. Of particular interest is the timing of the many reported effectors of the secretory vesicle Rab protein Sec4, including the myosin-V Myo2, the exocyst complex, the lgl homolog Sro7, and the small yeast-specific protein Mso1. At the trans-Golgi network (TGN) Sec4's GEF, Sec2, is recruited to Ypt31-positive compartments, quickly followed by Sec4 and Myo2 and vesicle formation. While transported to the bud tip, the entire exocyst complex, including Sec3, is assembled on to the vesicle. Before fusion, vesicles tether for 5 s, during which the vesicle retains the exocyst complex and stimulates lateral recruitment of Rho3 on the plasma membrane. Sec2 and Myo2 are rapidly lost, followed by recruitment of cytosolic Sro7, and finally the SM protein Sec1, which appears for just 2 s prior to fusion. Perturbation experiments reveal an ordered and robust series of events during tethering that provide insights into the function of Sec4 and effector exchange.

**\*For correspondence:**
RMG284@Cornell.edu (RMG);
apb5@cornell.edu (AB)

**Competing interest:** The authors declare that no competing interests exist.

## Editor's evaluation

This paper describes a tour de force characterization of the timeline and molecular events during the delivery and fusion of yeast secretory vesicles. It establishes a standard for further investigation of this important process.

## Introduction

Proteins destined for the secretory pathway are made on the endoplasmic reticulum, transferred to the Golgi complex, from where they are sorted into secretory vesicles that are transported and ultimately fuse with the plasma membrane to deliver their cargo. This pathway has been highly conserved in eukaryotes, from budding yeast to animal and plant cells. Indeed, many of the proteins involved in the secretory pathway were first identified and characterized in yeast (*Novick et al., 1980*; *Novick, 2014*), which remains the organism in which the secretory pathway is best understood. Despite several decades of research, during which the general function of the major proteins involved has been elucidated, many of the components have not yet been visualized in vivo at a spatiotemporal resolution sufficient to assess their order of action. A similar timeline has been previously defined for the events and components of yeast endocytosis, a process which involves hundreds of copies of some proteins and takes on the order of 10–12 s (*Picco et al., 2015*), but this work represents the first such timeline for exocytosis, the totality of which occurs over approximately 5 s.

This laboratory has previously analyzed and imaged the transport of budding yeast secretory vesicles, marked by the Rab Sec4, from the Golgi to the plasma membrane. Initial studies showed that

secretory vesicles are transported by the myosin-V motor protein, Myo2 along actin cables at about 3 µm/s (*Donovan and Bretscher, 2012*; *Donovan and Bretscher, 2015b*; *Santiago-Tirado et al., 2011*; *Schott et al., 1999*; *Schott et al., 2002*). In the current study, we sought to build on these results by imaging components involved in secretory vesicle biogenesis and exocytosis individually and in combination at rates significantly faster than previously achieved. Our goal was to generate a timeline along which the participation of each component could be recorded. Of particular interest is the timing of effectors of Sec4, the Rab protein which associates with secretory vesicles and has a number of known effectors, including the exocyst, the myosin-V motor Myo2, and Sro7 (*Guo et al., 1999*; *Jin et al., 2011*; *Rossi et al., 2018*; *Schott et al., 1999*). This goal presented a number of technical challenges. First, all components had to be tagged in such a way as to minimally impair their function when expressed from their cognate promoters. Second, the number of molecules involved for many of the components is very small, making single, and especially double-label, imaging challenging. Third, since events occur in the timeframe of seconds, rapid frame capture was imperative to allow imaging at a significantly higher rate and resolution than has so far been achieved.

In this study we were able to image secretory vesicle biogenesis, with the arrival at the trans Golgi network of the Rab GEF Sec2, the Rab Sec4, and its effector the myosin-V motor Myo2. During transport, the secretory vesicle recruits the exocyst, the Sec4-effector complex necessary for vesicle tethering. During tethering, Rho3 is recruited, followed by Sro7, and then very briefly by the SM protein complex, Sec1/Mso1. Perturbation experiments show that this time-line is remarkably robust to levels of these and other components. Finally, we show that the associated Sec1 and Mso1 have redundant membrane-recruitment domains that aid its surprisingly fleeting participation during exocytosis. This time-line, together with an estimation of the number of molecules of each component involved and their known functions (*Table 1*), provides a framework to better understand biogenesis of secretory vesicles and their consumption at the plasma membrane.

## Results

### Secretory vesicle formation and transport from the trans-Golgi network (TGN)

To capture secretory vesicle formation at the Golgi, we needed a marker for the *trans*-Golgi network (TGN). At the TGN, a GTPase cascade results in the generation of secretory vesicles (*Novick, 2016*). The Arf-GEF Sec7 activates Arf1, which in turn recruits Pik1-Frq1 (the phosphatidylinositol 4-kinase complex) and the TRAPPII Rab-GEF complex. TRAPPII then recruits Ypt31/32 (Rab11 homologs), which recruits another Rab-GEF Sec2, which finally activates and recruits the secretory vesicle Rab, Sec4 (*Thomas et al., 2019*; *Thomas and Fromme, 2016*; *Walch-Solimena et al., 1997*). Ypt31/32 and Sec7 have frequently been used as interchangeable TGN markers; however, Ypt31/32 level has recently been shown to peak on compartments a full eight seconds after Sec7 (*Highland et al., 2021*). While endogenously tagged Sec7-mNeonGreen occasionally appears to fragment into small compartments or vesicles, most Sec7 appears to dissipate around the time of fragmentation, leaving resulting vesicles difficult to identify (*Figure 1—figure supplement 1*). By contrast, mNeonGreen(mNG)-tagged Ypt31 can be followed through vesiculation and appears to remain on vesicles through fusion with the plasma membrane (*Figure 1A*). The population of Ypt31 which appears to remain on the plasma membrane is most likely an artifact of the fluorescent tag, though this was not probed further. For this reason, we assessed the timing of recruitment of Sec2, Sec4, and a Myo2-marker (*see Figure 1—figure supplement 2* for construction) to mScarlet-Ypt31 marked compartments.

All three components appeared to briefly colocalize with Ypt31 before breaking-off as a nascent secretory vesicle from the remaining portion of Ypt31, then moving diffusively, and finally being transported linearly towards the bud (*Figure 1A–E*). Sec2 first appeared on the Ypt31-containing TGN approximately 2 s prior to separation, whereas Sec4 and the Myo2-marker appeared approximately 1 s before separation, and, presumably, about one second after Sec2 (*Figure 1B–E*). The close temporal proximity of Sec4 and Myo2 arrival suggests that Myo2 is likely the first effector of Sec4 recruited to secretory vesicles. This early recruitment relative to other exocytic factors is probably aided by the fact that Myo2 is also a direct effector of Ypt31 (*Lipatova et al., 2008*). While Sec4, Sec2, and Myo2 were all found to be rapidly directed towards the bud following separation from the TGN—generally in under half a second—only compartments marked with Sec2 or Sec4 appeared capable of moving

**Table 1.** Proteins discussed in this work.

| Protein | Brief Description | Interactions Relevant to this Work |
|---|---|---|
| Ypt31/32 | Paralogous Rab proteins which localize to late Golgi to facilitate TGN transport by Myo2 and help drive vesicle formation. | Sec2, Myo2, Exocyst (Sec15) |
| Sec4 | The secretory vesicle Rab protein which recruits effectors critical to exocytosis. | Sec2, Myo2, Msb3/4, Exocyst (Sec15), Sro7 |
| Sec2 | GEF (Guanine-nucleotide Exchange Factor) for Sec4. | Ypt31/32, Sec4, Exocyst (Sec15) |
| Msb3/4 | Redundant GAPs (GTPase Accelerating Proteins) for Sec4. | Sec4 |
| Myo2 | Class V Myosin motor protein responsible for transporting secretory vesicles (among other cargo) to the growing bud. Effector of Ypt31 and Sec4. | Sec4, Exocyst (Sec15), Ypt31/32 |
| Smy1 | Kinesin-related protein which increases Myo2 affinity for Sec4. Does not function as a typical kinesin. | Myo2 |
| Exocyst | The hetero-octameric secretory vesicle tethering complex. | *See below:* |
| Sec3 | Aids in plasma membrane binding, SNARE assembly. Exocyst component. | Rho1, Cdc42, Sso1/2, PI(4,5)P$_2$ |
| Sec5 | Exocyst component. | N/a |
| Sec6 | Aids in SNARE assembly. Exocyst component. | Snc1/2, Sec1, Sec9 |
| Sec8 | Exocyst component. | N/a |
| Sec10 | Exocyst component. | N/a |
| Sec15 | Directs the exocyst to secretory vesicles. Exocyst component. | Ypt31/32, Sec2, Sec4, Myo2 |
| Exo70 | Aids in initial plasma membrane binding. Exocyst component. | Rho3, Cdc42, PI(4,5)P$_2$ |
| Exo84 | Aids in SNARE assembly. Exocyst component. | Sro7 |
| Rho3 | Polarity regulating Rho-GTPase. Helps "activate" the exocyst through Exo70. | Exocyst (Exo70) |
| Cdc42 | Polarity regulating Rho-GTPase. | Exocyst (Sec3, Exo70) |
| Snc1/2 | Secretory vesicle resident SNARE proteins. | Sso1/2, Sec9, Sec1, Mso1 |
| Sso1/2 | Plasma membrane resident SNARE proteins. | Snc1/2, Sec9, Sec1, Mso1 |
| Sec9 | Cytosolic SNARE protein. | Snc1/2, Sso1/2, Sro7, Sec6, Sec1, Mso1 |
| Sro7/77 | Lgl/Tomosyn homologs and effectors of Sec4. Aids in vesicle tethering and Sec9 regulation. | Sec4, Exocyst (Exo84), Sec9 |
| Sec1 | Sec1-Munc18 Family protein essential for secretory vesicle fusion with the plasma membrane. | Snc1/2, Sso1/2, Sec9, Mso1 |
| Mso1 | Small fungal Sec1 accessory which is thought to aid in SNARE assembly. Reported Sec4 effector. | Sec1, Sec4 |

diffusely for up to several seconds, suggesting that at least some vesicles can separate from the TGN without first acquiring Myo2.

It's remarkable that the machinery driving secretory vesicle formation has not yet been identified. The field has long held that since secretion itself is an essential process, formation of the classical

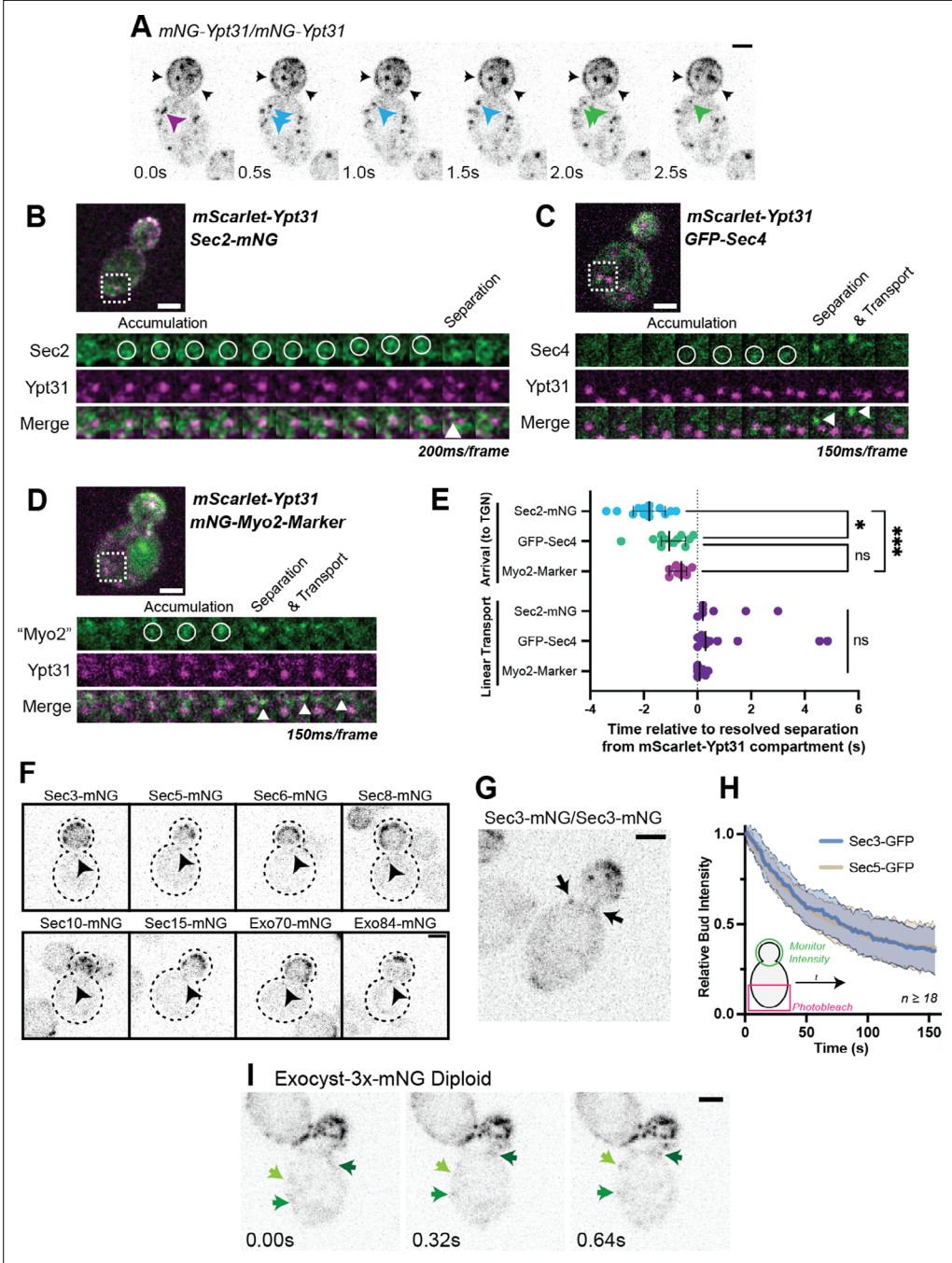

**Figure 1.** Recruitment of Sec2, Sec4, and Myo2 precede secretory vesicle budding from the TGN and all exocyst components are on secretory vesicles before arrival at the plasma membrane. (**A**) An example of mNG-Ypt31 TGN vesiculation in a diploid cell. Colored arrowheads follow successive fragmentation of an initial compartment. Black arrows highlight aberrant mNG-Ypt31 on the plasma membrane. (**B**) Timeseries example of Sec2-mNG recruitment to and budding from an mScarlet-Ypt31-marked compartment in a diploid cell. Single plane video. (**C**) Timeseries example of GFP-Sec4 recruitment to and budding from an mScarlet-Ypt31-marked compartment in a diploid cell. Single plane video. (**D**) Timeseries example of an mNG- Myo2-marker being recruited to and budding from an mScarlet-Ypt31-marked compartment in a diploid cell. Single plane video. (**E**) Order of recruitment to Ypt31 TGN compartments aligned by apparent separation of signal from the compartment. Budded vesicles were generally transported linearly towards the bud in under a second. Median ±95% CI. n≥10. *, p≤0.05; ***, p≤0.001. (**F**) Localization of mNG-tagged exocyst components in haploid cells. Sum projection of a 1.5 μm vertical volume surrounding the bud neck. Arrowheads indicate a single vesicle approaching the bud neck in each. All scaled

*Figure 1 continued on next page*

*Figure 1 continued*

equally; Bar on Exo84, 2 μm. (**G**) Additional example of mNG-Sec3 localizing to puncta approaching the bud neck in homozygously-tagged diploid cells. (**H**) Fluorescence Loss in Photobleaching (FLIP) experiment comparing the recycling of Sec3 or Sec5 from the bud. Mean curve ± SD, n≥18. (**I**) Exocyst-3x-mNG faintly localizes to vesicles earlier in the mother. Compare the moving signals identified by green arrows to any punctum within in the bud. All bars, 2 μm.

The online version of this article includes the following figure supplement(s) for figure 1:

**Figure supplement 1.** An example of apparent Sec7-mNG TGN fragmentation.

**Figure supplement 2.** Rationale for and construction of the Myo2 marker.

**Figure supplement 3.** Sec4 localizes to larger fusion-competent compartments in a *vps1*-null strain.

**Figure supplement 4.** While vesiculation dynamics of the underlying compartments appear similar, Ypt31 and Ypt32 are enriched on subtly different membranes marked by localization to either the mother cell or daughter bud.

**Figure supplement 5.** Construction of the exocyst marker and colocalization examples with other markers used in this study.

80–100 nm secretory vesicles must be carried out by similarly essential components. Progress in this regard has been at least partially hampered by the observation that there appears to be more than one redundant pathway ultimately leading to exocytosis (*Harsay and Bretscher, 1995*). Our preliminary experiments examining the non-essential Vps1, which has been implicated in a late step of the secretory pathway (*Gurunathan et al., 2002*; *Harsay and Schekman, 2007*), suggest that it is directly involved in the formation of at least one class of secretory vesicle. Loss of this small GTPase which is known to act at the endosome (*Chi et al., 2014*), results in cells containing larger compartments capable of maturing to the point that they recruit Sec4 and apparently fuse with the plasma membrane (*Figure 1—figure supplement 3*). This directly challenges the notion that the 80–100 nm vesicles themselves are essential for secretion, but will need to be examined more closely in future studies.

Ypt31 and Ypt32 are often described interchangeably as they are considered functionally redundant (deletion of one or the other is consistent with viability, whereas loss of both is lethal). Interestingly, while the relative recruitment timing of Sec2 and Sec4 was similar when compared to Ypt32 compartments (*Figure 1—figure supplement 4A*), relative Ypt32 to Ypt31 ratios appeared to mark subtly different compartments, with Ypt32 being more prominent on compartments within the bud (*Figure 1—figure supplement 4B* and C). This trend of compartments within the bud carrying a higher concentration of Ypt32 also held when the markers on Ypt31 and Ypt32 were reversed and only expressed from endogenous promoters (*Figure 1—figure supplement 4D*). Though Ypt31 and Ypt32 are paralogs generated during the whole genome duplication event and clearly provide similar-enough functions to compensate for the loss of one another, existing genetic data supports the possibility of the two proteins serving subtly different cellular functions. Of the 54 reported significant negative genetic interactors of Ypt32, only two overlap with significant negative genetic interactors of Ypt31 (of 182 total). Similarly, the two show no overlap in significant positive genetic interactions (*Costanzo et al., 2016*). An intriguing possibility is that Ypt32-enriched TGN within the bud could represent the more early-endosome-like TGN compartments with which endocytic cargo colocalize and seems consistent with earlier descriptions of post-Golgi endosomes in yeast and recycling endosomes in higher eukarya (*Day et al., 2018*; *Lewis et al., 2000*; *Pelham, 2002*).

We next examined the recruitment of the exocyst to secretory vesicles. The exocyst complex, comprised of single copies of Sec3, 5, 6, 8, 10, 15, Exo70, and Exo84, tethers secretory vesicles to the plasma membrane before fusion and is primarily thought to localize to the vesicles themselves, ahead of membrane delivery (*Boyd et al., 2004*). Sec3, however, has long been suggested to be a 'landmark' of secretory vesicle tethering—residing on the plasma membrane, apart from the other seven subunits of the complex on the vesicle itself. This long-standing model suggests that once a vesicle comes into proximity of the plasma membrane, Sec3 then joins the bulk of the complex to complete the exocyst and facilitate tethering (*Finger et al., 1998*; *Wiederkehr et al., 2003*). More recent studies suggest that the yeast exocyst complex is an obligatory heterooctamer, with Sec3 always being bound to the other components with no discernible subcomplexes (*Heider et al., 2016*).

In this study, each exocyst component, including Sec3, was individually tagged with mNeonGreen. When each component was imaged, they could be seen on vesicles in the mother cell, although clear vesicular localization was generally only noticeable in immediate proximity to the bud neck and within the bud itself (*Figure 1F and G*). Additionally, fluorescence loss of Sec3 and Sec5 from the bud during constant photobleaching ('FLIP') of the mother cell was identical, suggesting that Sec3 recycles to the mother cell at the same rate as other exocyst components (*Figure 1H*). Thus, Sec3 appears to be a component of the octameric exocyst complex in vivo which associates with secretory vesicles ahead of tethering and is not likely to be a spatial 'landmark' component of the exocyst. This echoes earlier work downplaying the landmark model of Sec3 function (*Roumanie et al., 2005*).

Localization of the exocyst to moving vesicles in the mother only became slightly more apparent upon multiply-tagging the complex with one copy of mNeonGreen on each of three subunits (Exocyst-3x-mNG; *Figure 1I*, *Figure 1—figure supplement 5A and B*). Within the mother, few examples of Sec4-Exocyst colocalization could be found, though the clearer examples occurred on vesicles which moved in a diffusive manner briefly before polarized movement (*Figure 1—figure supplement 5C*). While this further supports that our exocyst marker was identifying moving secretory vesicles, this

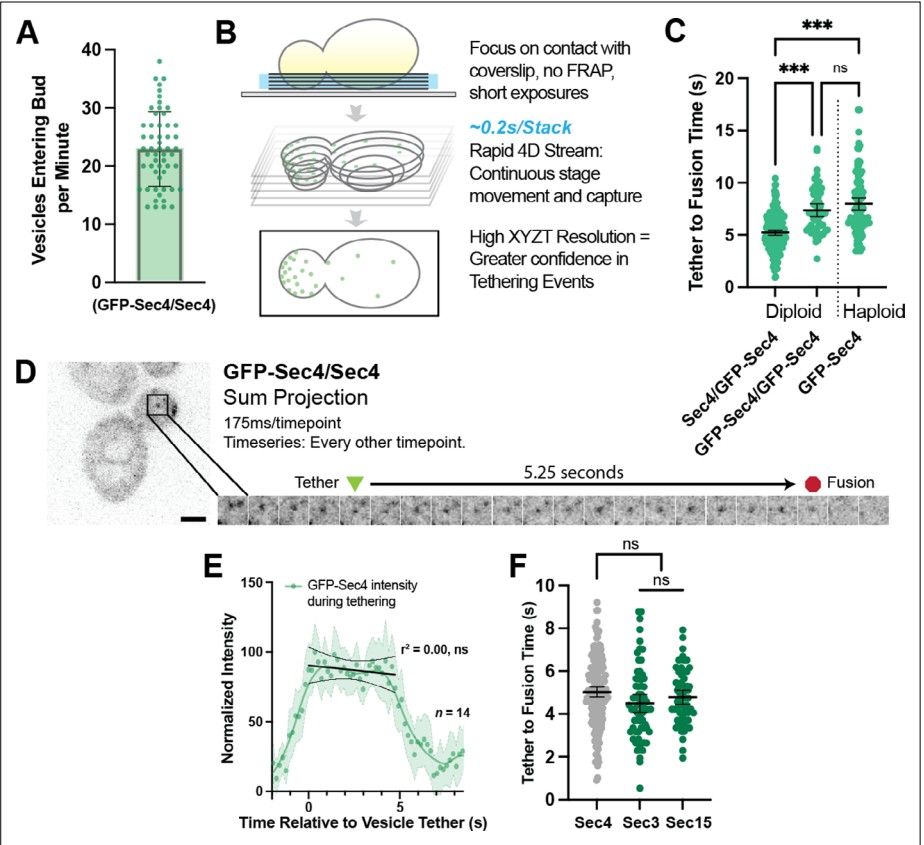

**Figure 2.** Secretory vesicles tether for about 5 s before fusion. (**A**) Approximately 22 vesicles enter the bud per minute. n=55 cells. Mean ± SD. (**B**) Schematic diagram of improved volumetric imaging technique used in this study. See *Video 1* for example. (**C**) Collected data of all timed GFP-Sec4 tethering events in wildtype GFP-Sec4/Sec4 cells as well as homozygously-tagged diploids and GFP-Sec4 haploids Means ±95% CI. GFP- Sec4/Sec4 Mean: 5.02 s, n=180 events, others n>50. ***, p≤0.001. (**D**) Example of GFP-Sec4 vesicle tethering and fusion from a GFP-Sec4 heterozygously-tagged diploid. Sum projection. See *Video 2*. (**E**) GFP-Sec4 fluorescence intensity is roughly constant through tethering. Local weighted regression (LOWESS; green) and linear regression curves (during tethering; black) added for visual interpretation. (**F**) Apparent tethering time of vesicles marked carrying Sec3-mNG (Mean: 4.55 s) or Sec15-mNG (Mean: 4.77 s) is similar to tethering time of vesicles marked by GFP-Sec4. Mean ±95% CI.

The online version of this article includes the following figure supplement(s) for figure 2:

**Figure supplement 1.** Multiple GFP-Sec4 vesicle tethering and fusion events can be visualized and timed in a single cell.

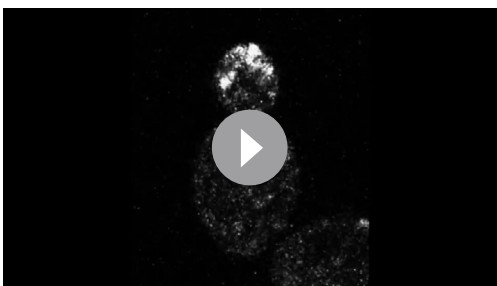

**Video 1.** Example of a typical three-dimensional live-cell vesicle tethering video showing a single diploid cell containing one endogenously-tagged copy of GFP-Sec4 over 26 seconds. Six planes were captured with 25ms exposures during continuous stage movement, resulting in 176ms per stack or 'timepoint'. Several vesicle fusion events, defined as rapid loss of GFP-Sec4 signal from stationary puncta, are highlighted with green arrows. Video is played back at 2 x capture speed.

https://elifesciences.org/articles/78750/figures#video1

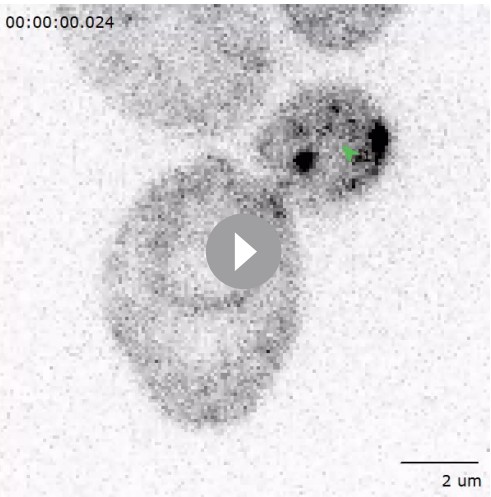

**Video 2.** Sum projection video of the vesicle tethering and fusion event illustrated in Figure 2D. Video is played back at 2 x capture speed.

https://elifesciences.org/articles/78750/figures#video2

rare observation stands in stark contrast to Sec4 or Sec2 which could be readily observed on vesicles at all points within the mother, indicating that exocyst association with vesicles generally occurs well after vesicle biogenesis and during transport.

## Improved techniques for the imaging of yeast exocytosis

Our previous study to examine secretory vesicle tethering utilized standard spinning disk confocal imaging of the cellular volume surrounding the bud neck of haploid *GFP-Sec4* cells (*Donovan and Bretscher, 2015a*). As most of the Sec4 in the cell is vesicle bound and in the bud, all the GFP signal in the bud was initially bleached and then incoming GFP-positive secretory vesicles were followed. An average of 22 vesicles enter the bud per minute (*Figure 2A*). The previous technique of intentionally photobleaching the bud thus necessarily relied on fortuitously timed secretory vesicle formation and transport from the mother and for that vesicle to then tether within the small vertical observation window centered around the bud neck. This technique confined critical information on vesicle tethering to the lowest resolution domain, the z-axis.

To overcome this limitation, we adjusted the imaged volume to the region of the cell proximal to the coverslip so that the majority of information regarding vesicle position was shifted into the higher resolution x- and y-axes (*Figure 2B*). Instead of photobleaching, we utilized the larger volume and plasma membrane surface area of mid-sized diploid buds to identify individual vesicles that tethered, fused, and could be confidently discerned from nearby vesicles. This three-dimensional capture was performed by streaming EM-CCD frames to RAM from a very small optically centered field of view, using short exposures (between 10 and 50ms), constant laser excitation, and continuous vertical movement of the stage. Additional discussion of the rationale for the imaging set-up used is provided in the Methods Section. By removing the need for discrete stage stepping, camera-shuttering, and laser firing this drove down the capture time per frame, resulting in typical captures of 175ms per stack of six planes for brighter signals from more abundant proteins, and less than a third of a second for very low abundance proteins. This is far faster than the capture of five planes in 1.5 s used in our earlier studies.

With these adjustments and careful analysis of these videos in 3D projections (*Video 1*), we were able to identify far more unambiguous tethering and fusion events than previously possible (*Figure 2C*, *Video 2*) and could even identify many events per cell in some captures (*Figure 2—figure supplement 1*). Here, we defined tethering as the first frame where a Sec4 vesicle came to a full stop at a location not more than one apparent vesicle diameter from where it eventually fused with the plasma membrane. Similarly, fusion was assumed to occur at the last frame in which signal could be positively identified at the tethering location. Although a fluorescent marker capable of visualizing cargo release

would be ideal for this task, to this date no such tool has been successfully developed. Analyzing many such events resulted in diploids resulted in a tether-to-fusion time of approximately five seconds (*Figure 2D*). Interestingly, we found early on in this study that the presence of the GFP-tag on Sec4 itself affects Sec4 function. A significant increase in tether-to-fusion time was seen when no untagged Sec4 was available (such as in a GFP-Sec4/GFP-Sec4 diploid or GFP-Sec4 haploid; *Figure 2C*). This difference indicated that the heterozygous diploid GFP-Sec4/Sec4 strain would be the most physi-ological framework in which to explore secretory vesicle tethering. By measuring the average inten-sity of GFP-Sec4 signal on tethered vesicles in the diploids, we found that GFP-Sec4 signal on the vesicle remains constant throughout the duration of tethering, only rapidly disappearing after several seconds, suggesting that Sec4 is not extracted from the vesicle during tethering (*Figure 2E*).

Next, we determined the time that exocyst components Sec3-mNG and Sec15-mNG associate with tethered vesicles. Both Sec3-mNG and Sec15-mNG remain on tethered vesicles for about the same time as the tether-to-fusion time of GFP-Sec4, although both consistently appeared slightly shorter (*Figure 2F*). Although this difference is not statistically significant, it is consistent with reports showing that the mammalian exocyst remains associated with exocytic vesicles roughly until fusion. Sec3 and Sec15 represent constituent components of the two putative exocyst sub-assemblies (Sec3, 5, 6, 8 vs 10, 15, Exo70, Exo84) and we do not see a significant difference in the residence timing of the two sub-complexes, as has been reported in mammalian cells, though this discrepancy could be a matter of confidence and sample size and underlying differences may be probed further in future exocyst-specific studies (*Ahmed et al., 2018*).

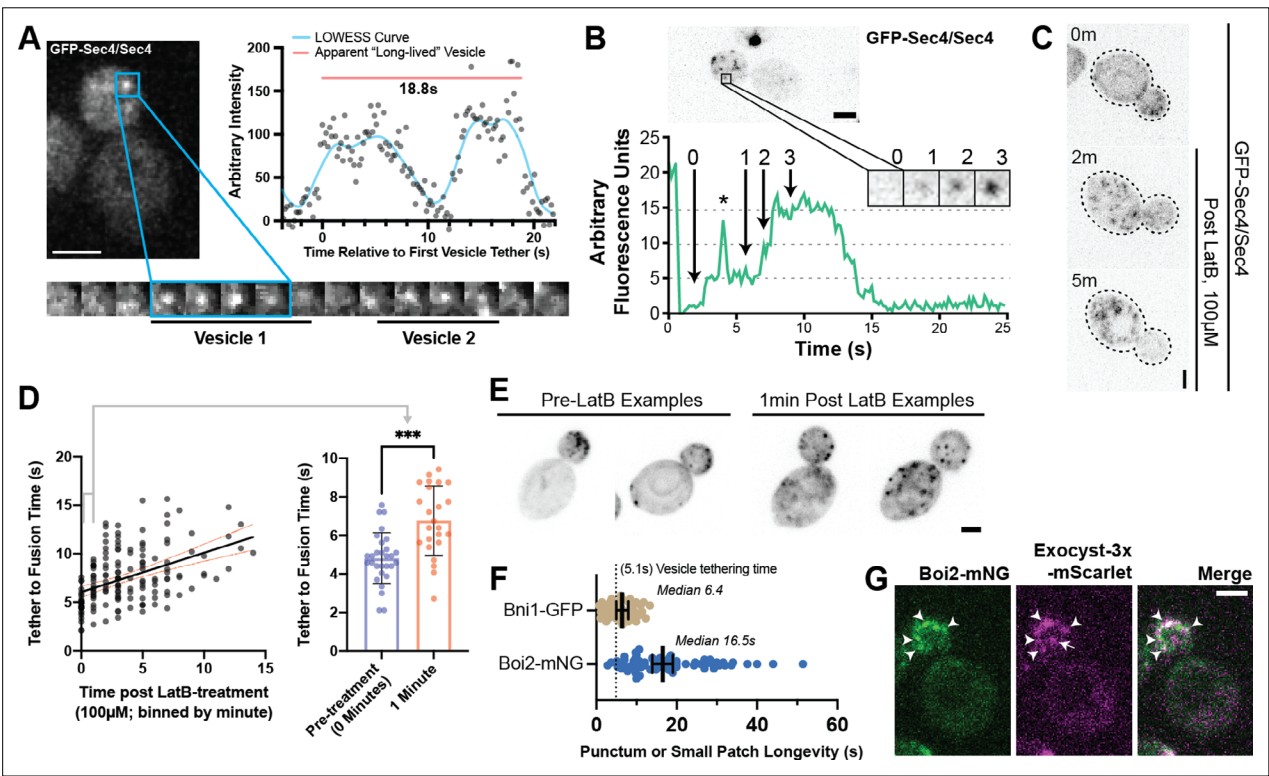

**Figure 3.** Tethering Hotspots Exist. (**A**) Sequentially tethering vesicles may appear as long-lived events in captures with low spatiotemporal resolution, especially when considering the elongated tethering time in cells with no untagged Sec4 (as in *Figure 2C*). Images were captured at 176ms per frame. Inset time-lapse shown with 10 x lower time resolution. (**B**) Additional example of "hot-spot" tethering shows 3 vesicles arriving and tethering in rapid succession at one un-resolvable location (see *Video 3*). Although they tether at separate times, they appear to fuse at roughly the same time. Asterisk marks signal from a bright vesicle that passed the observed position. (**C**) Secretory vesicle formation continues even in the absence of actin cables. (**D**) Disruption of actin cables with LatrunculinB (LatB) immediately results in elongated GFP-Sec4 vesicle tether-to-fusion time. ***, p≤0.001 by t-test. (**E**) Several examples of clustered GFP-Sec4 secretory vesicle localization in diploid cells not treated with LatB and examples of more dispersed vesicle tethering locations in cells 1 min after LatB treatment. Sum projection of cell bottom. (**F**) Boi2 patches are longer lived than individual vesicles or even Bni1 patches, suggesting a potential explanation for the observation of tethering 'hot-spots'. (**G**) Exocyst-3x-mScarlet frequently colocalizes with Boi2-mNG puncta on the plasma membrane. Arrowheads indicate vesicles colocalized with Boi2, arrow indicates a tethered vesicle not colocalized with Boi2.

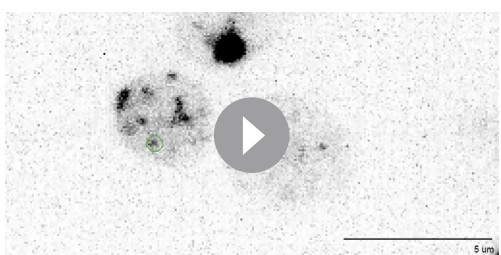

**Video 3.** Maximum intensity projection video of multiple vesicles tethering and apparently fusing at a single, un-resolvable location following a photobleaching event. Green circle illustrates the region from which intensity data was extracted in Figure 3B. Video is played back at approximately 3 x capture speed.

https://elifesciences.org/articles/78750/figures#video3

Earlier studies employing haploid GFP-Sec4 strains reported a longer tethering time of about 15–18 s (*Alfaro et al., 2011*; *Donovan and Bretscher, 2015a*), which cannot be explained by the use of a haploid GFP-Sec4 strain alone. During the current study, it became evident that a lower temporal resolution could result in longer apparent vesicle tethering due to a newly observed phenomenon of tethering 'hot-spots'. At these transient locations, secretory vesicles can be seen to tether sequentially (*Figure 3A*) and/or simultaneously (*Figure 3B*, *Video 3*). It is not currently possible to quantify the frequency with which this occurs due to the complexity of tracking the many vesicles within the bud for their entire pre-fusion lifetime. One possibility is that tethering hot-spots are near the end of actin cables that are used to transport vesicles to the plasma membrane. Disruption of actin cables with Latrunculin B resulted in an accumulation of secretory vesicles in the mother (*Figure 3C*) thereby precluding obtaining a definitive answer to the involvement of actin cables ends as hot-spots. Moreover, this disruption caused a significant elongation of tethering-to-fusion time with individual tethering events being moderately easier to identify due to their more broad distribution within the bud (*Figure 3D and E*). Interestingly, though puncta and patches of Bni1 (the formin within midsized buds responsible for actin cable assembly) only persist for

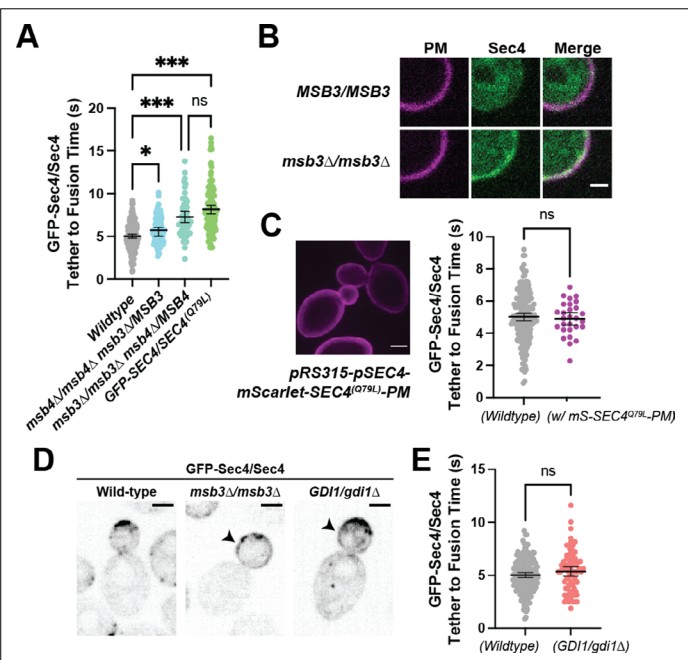

**Figure 4.** Duration of vesicle tethering is modulated by the level of Sec4:GTP on the vesicle. Msb3 likely acts on Sec4 twice to aid in efficient tethering and Sec4 recycling. (**A**) Homozygous deletion of msb3 or replacement of one copy of Sec4 with a constitutively active allele results in significantly longer tether-to-fusion time, whereas similar deletion of msb4 has a much milder effect. *, p≤0.05; ***, p≤0.001. (**B**) Deletion of msb3 results in GFP-Sec4 accumulation on the plasma membrane (PM). Unbudded cells shown for simplicity, though the same occurs in budded cells. PM is marked with mCherry-Ist2tail(2 X). Bar, 1 µm. (**C**) Expression of a constitutively active and plasma-membrane-bound Sec4 has no effect on GFP-Sec4 vesicle tethering time. Ectopic Sec4:GTP on the plasma membrane is not the cause of elongated tethering times observed in msb3Δ and Sec4(Q79L). (**D**) Heterozygous deletion of Gdi1 (an essential protein) also induces Sec4 accumulation on the plasma membrane. (**E**) Heterozygous deletion of Gdi1 has no significant effect on secretory vesicle tether-to-fusion time.

slightly longer than the lifetime of a single vesicle tethering event (mean, 6.4 s), patches of another exocytic protein, Boi2, were capable of lasting significantly longer (median, 16.5 s; *Figure 3F*). The redundant Boi1/2 proteins have been suggested to play a role in both actin and vesicle tethering regulation (*Glomb et al., 2020*; *Masgrau et al., 2017*) and, indeed many, but not all, vesicles appear to tether at Boi2-mNG puncta (*Figure 3G*).While this could potentially explain the underlying biology of tethering hot-spots, the actual mechanisms and dynamics of tethering hot-spots will need to be explored in a future study.

## Modulation of tethering by Sec4-GTP state

Sec4 GTP-hydrolysis, as promoted by the genetically redundant Rab-GAPs Msb3 and Msb4, is believed to be important for maintaining proper tether-to-fusion time (*Donovan and Bretscher, 2015a*). Homozygous deletion of *msb4* with heterozygous deletion of *msb3* (*msb3Δ/MSB3 msb4Δ/ msb4Δ*) leads to a modest but significant increase in tether-to-fusion time, while the opposite (*msb3Δ/ msb3Δ msb4Δ/MSB4*) results in much greater increase in tether-to-fusion time. A similar effect on tethering can be seen when one copy of Sec4 in a diploid is replaced with the constitutively active *Sec4$^{(Q79L)}$* allele (*Figure 4A*). Excess active Sec4 is seen on the plasma membrane of *msb3Δ/msb3Δ* cells (*Figure 4B*). Thus, the longer tethering times observed could be directly due to delayed hydrolysis of Sec4-GTP on the vesicle, or indirectly due to excess active Sec4 on the plasma membrane which may be sequestering secretory effectors, limiting their availability for tethered vesicles, and thereby elongating tethering time. To distinguish between these, we expressed a constitutively active *mScarlet-Sec4$^{(Q79L)}$-Ist2$^{tail}$* (aka. *mScarlet-Sec4$^{(Q79L)}$-PM*) which embeds into the plasma membrane (*Figure 4C*, left). Notably, there was no difference in vesicle tether-to-fusion time with expression of the constitutively active, plasma membrane-bound Sec4 (*Figure 4C*, right). Thus, delayed hydrolysis of vesicle-bound Sec4-GTP elongates tethering time. These results also highlight the inherent context-dependent regulation of exocytic factors and supports the idea that local coincidence detection mechanisms are important for the function of Sec4 effectors.

Limitation of GDI availability by heterozygous deletion of *Gdi1* also had no significant effect on vesicle tether-to-fusion time despite also resulting in accumulated Sec4 on the plasma membrane (*Figure 4D and E*). In addition to further supporting the conclusion that accumulated Sec4 on the plasma membrane does not interfere or elongate tether-to-fusion time, the fact that Sec4 can be retained in this way strongly supports that the rapid loss of Sec4 signal from a tethered vesicle is representative of fusion.

## Defining the location and timing of components individually

We next sought to examine how long other components involved in exocytosis were associated with tethered vesicles. To prevent potential perturbation of tethering by over-expression, single components were homozygously tagged in the genome and expressed exclusively under their own promoters. Components that were observed to reside on vesicles during transport were timed, much like with Sec4 itself, from the moment of tethering until punctate signal was lost. Components that appeared to only form stable punctate structures at the plasma membrane (i.e. not on moving vesicles) were timed from the first frame the punctum appeared to the last frame it was visible.

Unlike Sec4 and the exocyst complex, Sec2 and Smy1 appear to dissociate from secretory vesicles shortly after tethering (*Figure 5A*). Sec2 is the first protein observed to depart the vesicle, with Sec2-mNG puncta lasting on average 2.8 s after becoming stationary at the plasma membrane. Smy1, which associates with vesicle-bound Myo2 and resides on vesicles through transport (*Lillie and Brown, 1994*; *Lillie and Brown, 1992*; *Lwin et al., 2016*), only remains associated with the vesicle for an average of 3 s after tethering. Since tagged wildtype Myo2 cannot be observed due to a high cytoplasmic background of unactivated Myo2, Smy1 likely parallels Myo2 behavior. Indeed, assessing the time of punctate signal loss of the Tomato-Myo2-marker supports that Smy1 and Myo2 dissociate around the same time in vesicle tethering (*Figure 5—figure supplement 1*).

Three additional exocytic components, Sro7, Mso1, and Sec1, were found to exclusively localize to puncta at the plasma membrane. Sro7 is a nonessential protein involved in polarity maintenance and secretion that directly binds and regulates the soluble SNAP-25 homolog, Sec9 (*Hattendorf et al., 2007*; *Lehman et al., 1999*). Earlier studies have suggested that Sro7 localizes broadly to the plasma membrane, however, this was shown via immunofluorescence following 2µ overexpression (*Lehman*

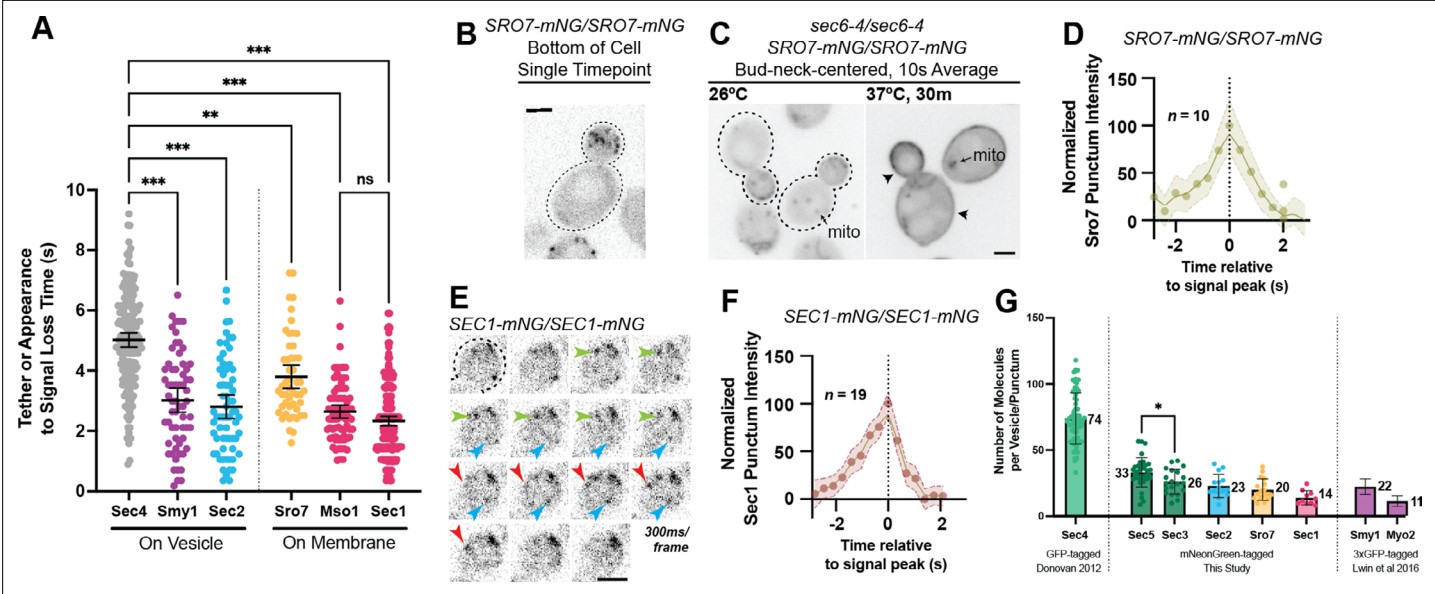

**Figure 5.** Defining the location and timing of exocytic components individually. (**A**) Independent timing from tether to disappearance for components residing on vesicles (Smy1-mNG, Sec2-mNG) and timing of punctum appearance to disappearance for components apparently residing on the plasma membrane (Sro7- mNG, Sec1-mNG, Mso1-mNG). All imaged in homozygously-tagged diploids in a manner as in *Figure 2B*. **, p≤0.005; ***, p≤0.001. See *Figure 5—figure supplement 2* for combined statistics. (**B**) Sro7, a Sec4 effector, does not localize directly to vesicles and instead appears in short-lived puncta at the plasma membrane. Sum projection. Bar, 2 µm. (**C**) Disruption of secretory vesicle tethering via sec6-4 does not result in Sro7 accumulation on cytosolic vesicles, but instead accumulation at the plasma membrane. Mitochondrial autofluorescence is apparent due to the low signal intensity of Sro7-mNG. Bar, 2 µm. (**D**) Averaging multiple events shows that the timing of arrival and departure for Sro7 is roughly symmetrical. (**E**) Time-lapse of several Sec1-mNG puncta within the bud. Sum Projection. Captured as in *Figure 2B*. ~300ms per frame. Bar, 2 µm. (**F**) Averaging several Sec1-mNG localization events shows faster signal dissipation than accumulation. This is different from what is observed for another PM-localized protein Sro7 in (**C**). (**G**) Collected data of number of molecules per vesicle of various components measured both in this study and other studies from our lab. *, p≤0.05.

The online version of this article includes the following figure supplement(s) for figure 5:

**Figure supplement 1.** A Myo2-marker co-imaged with GFP-Sec4 indicates loss of Myo2 occurs midway through vesicle tethering.

**Figure supplement 2.** Combined data from *Figures 2F and 5A*.

*et al., 1999*). Imaging of Sro7-mNG expressed from its own promoter, shows that Sro7 does, indeed, localize to the plasma membrane, however, it does so in a polarized and punctate pattern similar to tethered vesicle localization (*Figure 5B*). The median longevity of Sro7 puncta was approximately 3.4 s (mean, 3.8 s; *Figure 5A*).

None of the Sro7 signal appeared to localize to diffusive or actively transported secretory vesicles, whether in the mother or bud. This non-vesicular localization was somewhat surprising as the redundant Sro7 and Sro77 are thought to be direct effectors of Sec4:GTP (*Rossi et al., 2018*; *Watson et al., 2015*). Additionally, disruption of vesicle tethering by a *sec6-4* mutation induced broad Sro7-mNG localization across the plasma membrane (*Figure 5C*), perhaps due to its known association with the exocytic SNARE Sec9. This result is also surprising as Sro7 has been reported to induce secretory vesicle clustering, at least when overexpressed (*Rossi et al., 2020*). The arrival and departure kinetics of Sro7, as measured by average punctum intensity over time, were roughly symmetrical, with the longevity before peak intensity being about equal to the longevity after the peak (*Figure 5D*).

Sec1, the SM (Sec1-Munc18) family protein responsible for directing secretory vesicle SNARE assembly, has primarily been visualized at low spatiotemporal resolution or via BiFC (*Carr et al., 1999*; *Kustermann et al., 2017*; *Weber et al., 2010*). Here, we visualized Sec1-mNG (*Figure 5E*, *Video 4*) and found the shortest localization timing of any exocytic component examined, with Sec1 puncta longevity having a median time of 2.1 s (mean, 2.3 s; *Figure 5A*). Interestingly, unlike Sro7, Sec1 arrival and departure times were different, with the accumulation of signal taking slightly longer than its dissipation following peak intensity (*Figure 5F*). This is potentially consistent with Sec1 functioning at the end of vesicle tethering, rapidly disappearing around the moment of membrane fusion.

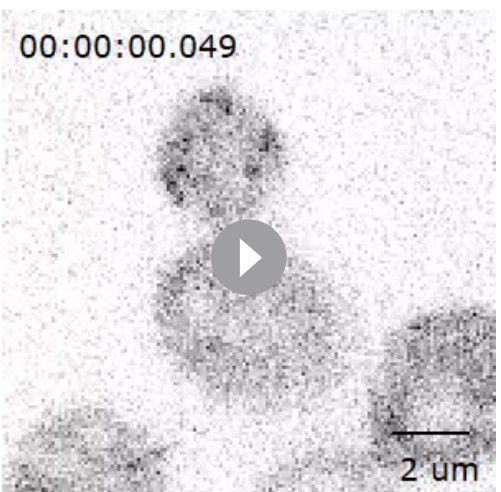

**Video 4.** Sum projection video of Sec1 localization events in a Sec1-mNG homozygous diploid cell. Six planes were captured with 50ms exposure times, resulting in 350ms per timepoint. For ease of visibility, video is shown using a two-frame rolling signal average. Video is played back at 4 x capture speed.

https://elifesciences.org/articles/78750/figures#video4

Mso1, a small nonessential protein tightly associated with Sec1 and suggested to be an effector of Sec4, was found to have a similarly short lifetime, not significantly different from Sec1 (median 2.5 s, mean 2.6 s; *Figure 5A*; *Weber et al., 2010*; *Weber-Boyvat et al., 2013*).

To estimate the relative number of molecules associated with secretory vesicles, we determined the maximum fluorescence intensity, compared with the known standard Cse4-mNG (*Lawrimore et al., 2011*), of several components which had not been previously quantified, and added it to data from previous publications (*Figure 5G*). The exocyst complex has previously been reported to localize with approximately 15 complexes per vesicle (*Picco et al., 2017*) and this analysis, while done utilizing a different technique and different fluorescent tags, showed roughly similar results (26±9 and 33±11 respectively). About 23±9 molecules of Sec2 could be seen on vesicles during transport. Finally, few copies of Sro7 (20±8) and Sec1 (14±5) were found to localize to their respective membrane puncta in the bud.

## Constructing a timeline

Having measured the timing of various components individually, we next sought to correlate these timings into a cohesive timeline of events from initial tethering to fusion. This endeavor was a complex one. Most of the proteins examined have a low number of molecules associated with each tethering event, so ensuring sufficient fluorescent intensity for meaningful detection of multiple markers, each expressed at endogenous levels, while maintaining a reasonable imaging frequency, was challenging. This was further complicated by synthetic negative genetic interactions encountered between some fluorescently-tagged protein pairs (such as diminished growth or perturbed protein localizations compared to singly-tagged strains), potentially due to induced steric clashes. Additionally, even the most abundant vesicular protein, Sec4, was not bright enough for our experiments when tagged heterozygously with mScarlet, the brightest currently available red fluorescent protein.

Unfortunately, GFP-Sec4 paired poorly with our best red vesicle marker Exocyst-3x-mScarlet, with tethering time as measured by GFP-Sec4 alone in this strain being somewhat elongated (*Figure 6—figure supplement 1*). However, even in this strain we can see that Sec4 and the exocyst depart from exocytic sites around the same time (*Figure 6A*). Fortunately, Exocyst-3x-mScarlet was able to be successfully utilized in combination with several other tagged components with minimal detriment. Further, since several components (Sec2, Myo2, Smy1, and the exocyst itself) are transported to the tethering site on the vesicle, alignment of loss of these components to the moment of tethering is easy, for example tracking the loss of a Tomato-Myo2-marker compared to GFP-Sec4, confirms that Myo2 begins dissociating shortly after the start of tethering (*Figure 6B*; *Figure 5—figure supplement 1*).

One of the first non-vesicular components to colocalize with tethered vesicles after their arrival to the plasma membrane is Rho3. Although Rho3 plays an integral role in vesicle tethering, it has not been shown to be present on constitutive secretory vesicles themselves (*Forsmark et al., 2011*; *Robinson et al., 1999*). Rather, in wildtype cells, Rho3 is broadly resident on the plasma membrane with transient bud-localized patches of increased concentration (*Figure 6—figure supplement 2*). Rho3 was visualized with a recently developed internal-mNG (imNG) tag and compared to the arrival of vesicles marked by Exocyst-3x-mScarlet (*Figure 6C*; *Gingras et al., 2020*). Alignment of multiple such events illustrates that Rho3 peaks at sites of tethered vesicles about 1.5 s after tethering (*Figure 6D*).

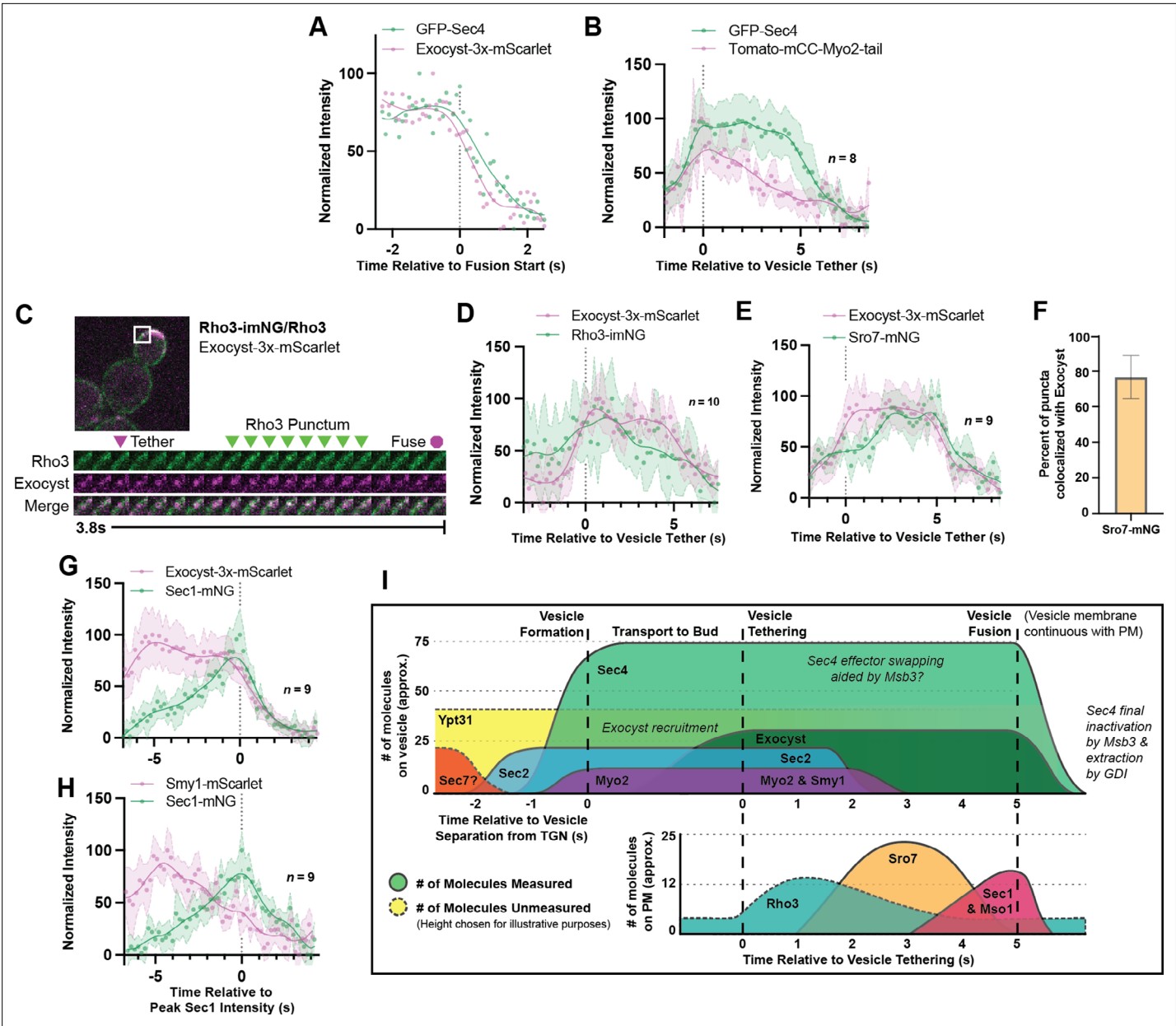

**Figure 6.** Relative ordering of exocytic events and the tether-to-fusion timeline. (**A**) Single example of vesicle fusion showing near simultaneous loss of Sec4 signal and the exocyst in GFP-Sec4/Sec4 Exocyst-3x- mScarlet. (**B**) Averaging of several Sec4 vesicle tethering events shows that on average Myo2 begins dissociating from the vesicle around the start of tethering. (**C**) Internally tagged Rho3-imNG on the plasma membrane concentrates briefly after Exocyst-3x-mScarlet vesicle tethering. (**D**) Averaging several vesicle tethering events as in C illustrates that Rho3-imNG membrane intensity rises with Exocyst-3x-mScarlet arrival and peaks ~1 s after vesicle tethering. Aligned by visual start of tethering. (**E**) Averaging several Exocyst-3x-mScarlet vesicle tethering events shows that Sro7-mNG localization to vesicles peaks 3–4 s after tethering. Aligned by visual start of tethering. (**F**) Not all Sro7-mNG puncta clearly colocalize with exocyst- marked secretory vesicles. The fraction of Sro7-mNG puncta visually colocalized with the exocyst in still images was manually counted for >10 cells across three biological replicates. Mean ± SD. (**G**) Averaging of several vesicle tethering events shows colocalization of Sec1-mNG and Exoy4st-3x-mScarlet with Sec1 signal peaking around the moment of exocyst loss. Aligned by moment of peak Sec1 intensity. (**H**) The start of Smy1- mScarlet loss from vesicles occurs approximately 5 s before Sec1-mNG peak. Aligned by moment of peak Sec1 intensity. (**I**) Timeline of events from secretory vesicle formation to plasma membrane fusion. Timing of appearance and disappearance of proteins in this timeline is based on the individual component data (where available) and aligned with the dual component imaging data.

The online version of this article includes the following figure supplement(s) for figure 6:

**Figure supplement 1.** Tether-to-fusion timing of GFP-Sec4 in Exocyst-3x-mScarlet diploid cells.

**Figure supplement 2.** Example of Rho3-imNG on its own in a haploid cell.

When Sro7-mNG was imaged alongside Exocyst-3x-mScarlet, Sro7 signal peaked between 3 and 4 s after tethering (*Figure 6E*). While Sro7 signal appears to plateau and remain longer than expected in this context when compared to imaging of Sro7 alone (*see Figure 5D*), this is likely an artifact of variable Sro7 arrival time. Regardless, it is evident that Sro7 localizes to sites of exocytic vesicles about 2 s after tethering and leaves around the moment of fusion. Interestingly, while Sro7 almost always co-localized with clear exocyst puncta in still images (as expected by the timing of its arrival relative to tethering), approximately 20% of Sro7-mNG puncta did not colocalize (*Figure 6F*). Although it's not clear what these Sro7-only puncta represent, it is possible that there is simply an undetectable quantity of exocyst complexes residing on the associated vesicle. Alternatively, these may represent the remnants of aborted tethering events, or more intriguingly, events with a class of vesicle more highly dependent on Sro7 for fusion, such as the ones suggested might contain proteins required for salt-stress tolerance (*Forsmark et al., 2011*).

Alignment of several Sec1 puncta co-imaged with Exocyst-3x-mScarlet showed that peak Sec1 intensity corresponded with the moment of exocyst loss, the expected moment of vesicle fusion (*Figure 6G*). When the timing of this Sec1-mNG peak was compared to Smy1-mScarlet, we found that Smy1 arrived and began decreasing in intensity (like Myo2 in *Figure 6B*) approximately 5 s before Sec1 peak, indirectly illustrating the time between vesicle arrival and fusion (*Figure 6H*).

Together, we can generate a timeline of events from initial secretory vesicle arrival and tethering to terminal fusion with the plasma-membrane (*Figure 6I*). The data paint a picture where secretory vesicles are delivered to the bud tip along actin cables and tethering is aided by the directive force of Myo2 as loss of actin cables extends tethering time. Shortly after arrival, the Sec4 GEF, Sec2, dissociates from the vesicle, followed closely by release of Myo2 and its cofactor Smy1. During this period, Rho3 on the plasma-membrane begins to associate with the tethered vesicle, likely enforcing tethering through its interactions with exocyst (*Adamo et al., 1999*; *Robinson et al., 1999*). Once the exocyst is activated and adopts an "open" conformation thanks to interactions with Rho proteins and the membrane itself (*Rossi et al., 2020*), Sro7 is recruited to the putative site of exocytosis by inter-actions with newly unoccupied Sec4:GTP molecules on the vesicle and the exocyst itself. Finally, with the chaperoned recruitment of Sec9 by Sro7, Sec1/Mso1 begins to concentrate around the vesicle, aided by direct interactions with Sec6 of the exocyst (*Morgera et al., 2012*). Sec1 then templates and stabilizes trans-SNARE complex assembly before rapid fusion, dissociation of the exocyst, and eventual extraction of Sec4 from the plasma membrane.

## Tether to fusion time is remarkably robust

With a newly defined timing from secretory vesicle tethering to fusion, we wished to identify which components regulated this timing by examining how the reduction or over-expression of various components affected tethering time. To accomplish this, individual components were either overex-pressed via multicopy 2μ plasmids or deleted from the genome (heterozygously for essential proteins, homozygously for nonessential). Surprisingly, limitation or overexpression of few proteins appeared to significantly alter vesicle tether-to-fusion time.

The mechanics of vesicle fusion is driven by the SNARE proteins, schematically shown in *Figure 7A*, so we first examined the effect of varying their levels on tethering time. Loss of the plasma membrane SNAREs Sso2 or heterozygous reduction in the functionally redundant Sso1, resulted in a longer teth-ering time, as did the heterozygous reduction of Sec9. Heterozygous reduction in the vesicle SNARE Snc2, which is functionally redundant with Snc1, had little effect (homozygous loss of Snc2 results in vesicle accumulation which precluded measurement of tethering time). While SNARE limitation had a clear detrimental effect on tether-to-fusion time, over-expression of either Snc2, Sec9, or Sso2 had a consistently minor, but not significant, effect of lowering tether-to-fusion time (*Figure 7B*). Thus, the role of SNAREs appears to be largely regulated by mass action, with their wildtype levels in modest excess.

Next, we examined proteins that regulate the level of active Sec4, the vesicle-bound Rab GEF Sec2, and the plasma membrane bound Rab GAP Msb3. While Msb3 deletion was shown to elon-gate vesicle tethering time (*see Figure 4A*), its overexpression had no such clear effect. Neither did overexpression nor heterozygous reduction of Sec2 (*Figure 7C*). It is surprising that over-expression of Sec2, which should elevate Sec4:GTP on vesicles, has no effect on tethering, whereas loss of Msb3 (*msb3Δ/msb3Δ*), that should also elevate Sec4:GTP, elongates tethering. To see where in the time-line

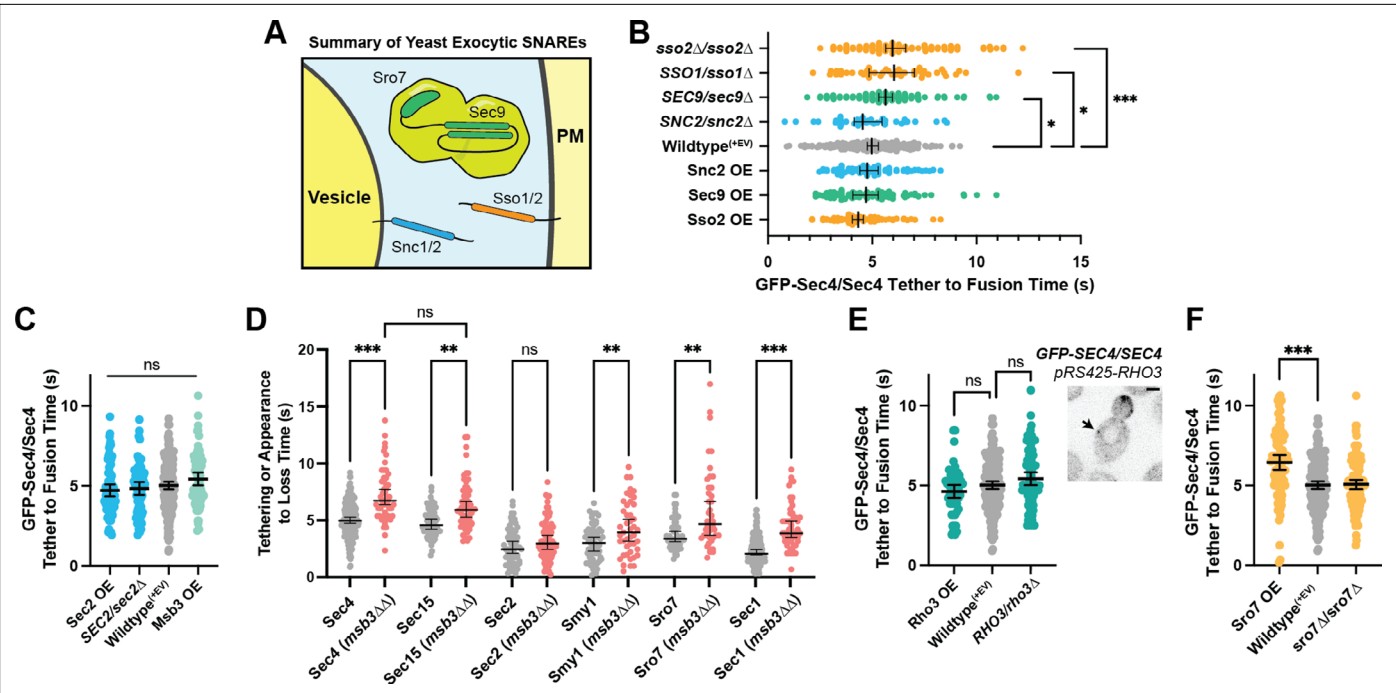

**Figure 7.** Secretion is remarkably robust.

(**A**) Expanded summary diagram of exocytic SNARE localization. The cytosolic SNARE Sec9 is shown bound to Sro7. (**B**) SNARE limitation significantly elongates vesicle tether-to-fusion time, while overexpression has the minimal, but statistically insignificant effect of decreasing tether-to-fusion time. Median ±95%CI. (**C**) Overexpression of Sec2, Msb3 (or heterozygous deletion of Sec2) has no significant effect on secretory vesicle tether-to-fusion time. Mean ±95% CI. (**D**) In msb3ΔΔ msb4Δ diploid cells, all components measured, except Sec2, remain significantly longer on tethered vesicles/plasma- membrane. All but Sec4 were homozygously tagged with mNG. Individual comparisons performed via Mann- Whitney test. Median ±95% CI. (**E**) Overexpression or heterozygous deletion of Rho3 has no significant effect on secretory vesicle tether-to-fusion time in the bud, however, aberrant, longer lived vesicle tethering events could be found in the mother cell when Rho3 was overexpressed. Mean ±95%CI. (**F**) Overexpression of Sro7 significantly elongates vesicle tether-to-fusion time. Mean ±95% CI. All panels: Wildype(+EV) shown for visual clarity. All overexpressions were compared to a relevant empty vector control, while deletions were compared the prior wildtype vesicle tethering data. See *Figure 7—figure supplement 1* for complete data and methods for more details. *, p≤0.05; **, p≤0.005; ***, p≤0.001.

The online version of this article includes the following figure supplement(s) for figure 7:

**Figure supplement 1.** All GFP-Sec4/Sec4 vesicle tether-to-fusion times; deletions and over expression comparisons and controls.

this extension of tethering occurred, we examined the duration of individual components in *msb3Δ/msb3Δ* cells compared with wildtype. This reveals that the rapid release of Sec2 from secretory vesicles after tethering is unchanged, whereas loss of Myo2 and the exocyst is delayed and the duration of Sro7 and Sec1 is extended (*Figure 7D*). Thus, reducing the hydrolysis rate of Sec4:GTP is uncoupled from the release of Sec2, but affects all downstream events.

Rho3, which likely plays an important role in the initial establishment of tethering, also had no significant effect on tether-to-fusion time when either overexpressed or heterozygously deleted (*Figure 7E*). Despite this, when *Rho3* was overexpressed ectopic vesicle tethering could occasionally be observed within the mother (*Figure 7E*, inset), though these tethering events were largely non-productive. Consistent with its non-essentiality, deletion of both copies of Sro7 showed no significant effect on vesicle tethering time (*Figure 7F*), whereas overexpression of Sro7 via a 2μ plasmid, resulted in a significant elongation of vesicle tethering time. This is best explained as an effect caused by diluting Sec9 at the tethered vesicle, as heterozygous deletion of Sec9 also resulted in a significant increase in tethering time (*Figure 7B*). Since there are relatively few Sec9 molecules in the cell, a gross overabundance of Sro7 (from overexpression) results in Sro7 being recruited to vesicles without bringing along the SNARE it's responsible for chaperoning (*Hattendorf et al., 2007*; *Lehman et al., 1999*; *Watson et al., 2015*).

Overexpression of another key Rho protein, Cdc42, as well as other regulatory components also had no significant effect on tethering time (*Figure 7—figure supplement 1B, C*). Notably, under

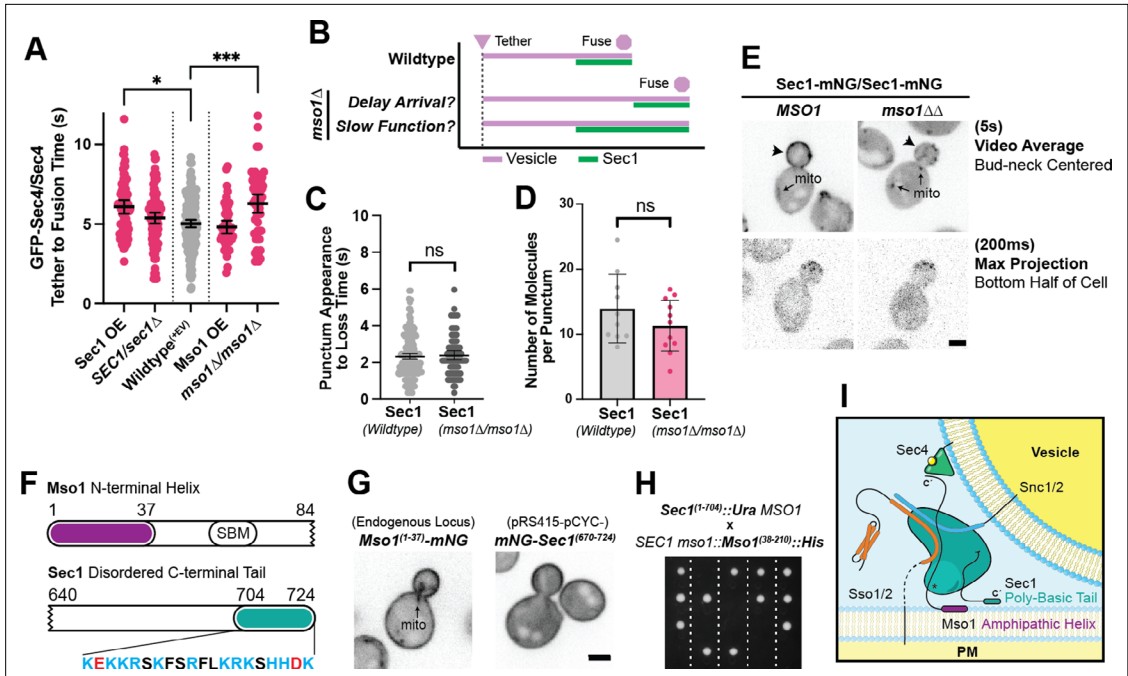

**Figure 8.** Sec1 and Mso1 both contribute to Sec1 membrane recruitment and function. (**A**) Sec1 overexpression and deletion of mso1 significantly elongate vesicle tether-to-fusion time, while heterozygous deletion of Sec1 has little effect. *, p≤0.05; ***, p≤0.001. (**B**) Two potential models for elongation of vesicle tethering induced by loss of mso1. Mean ±95%CI. (**C**) Sec1-mNG puncta have similar longevity to wildtype in mso1Δ cells. Mean ± SD. (**D**) The number of Sec1 molecules per membrane punctum is unchanged in mso1Δ cells. One-tailed unpaired t-test, n≥10, p=0.1. (**E**) Broad plasma-membrane association of Sec1 is diminished in mid-sized mso1Δ cells. (**F**) Schematic diagram of Mso1 N-terminal amphipathic helix and Sec1-Binding Motif (SBM) and the sequence of a portion of the Sec1 C-terminal tail. (**G**) The amphipathic alpha-helical N- terminus of Mso1 (aa1-37) and the C-terminus of Sec1 (aa670-724) both aid in plasma membrane localization. (**H**) Tetrad dissections show that loss of both the Mso1 N-terminus (aa1-37) and the last 20 residues of the Sec1 C-terminus is synthetically lethal. Five representative dissections are shown. See *Figure 8—figure supplement 1* for controls. (**I**) With initial localization aid via Mso1, Sec1 templates the assembly of trans-SNARE complexes. A theoretical, but likely, intermediate state with Sec1 simultaneously bound to Sso1/2 and Snc1/2 is shown (*Baker et al., 2015*). Mso1's N-terminus binds to the plasma membrane and interacts with Sec1 through its Sec1-binding motif (*) while Sec1 also interacts directly with the plasma membrane through its poly-basic tail. Loss of both of these PM-binding motifs is lethal. Mso1 may also contribute through reported interactions with the SNAREs and its C-terminus (**C'**) may interact with Sec4 to aid in recruitment of the complex to tethered vesicles (*Weber et al., 2010*; *Weber-Boyvat et al., 2011*).

The online version of this article includes the following figure supplement(s) for figure 8:

**Figure supplement 1.** Controls and complete results for Sec1 truncation and replacement of mso1 for tetrad dissection experiments.

**Figure supplement 2.** Alignment of many fungal Sec1 tails.

**Figure supplement 3.** Alignment of several Mso1 protein sequences of select ascomycetes from *Figure 8—figure supplement 2*.

the stringent statistical tests required for the many comparisons made in this study, the only protein for which overexpression resulted in a statistically significant lower vesicle tether-to-fusion time was Myo2. Together with the observation that loss of actin cables lengthens tethering times (*Figure 3C*), this result suggests that F-actin and Myo2 participate in the establishment of productive tethering (*Figure 7—figure supplement 1B*).

## Mso1 works with Sec1 to aid Sec1 localization and facilitate efficient exocytosis

Overexpression of Sec1 resulted in a slightly elongated tether-to-fusion time (*Figure 8A*). The under-lying mechanism of this increase may be similar to that of the response to Sro7 overexpression: Sec1 has little to no affinity for single SNAREs and binary SNARE complexes under normal conditions, but excess Sec1 may stabilize non-productive Sec1-SNARE complexes (*Carr et al., 1999*; *Hashizume et al., 2009*; *Togneri et al., 2006*). Heterozygous deletion of Sec1 had no clear effect on tether-to-fusion time. Unsurprisingly, overexpression of the nonessential Mso1 protein also had no measurable

effect, however, a strain lacking this same protein exhibited one of the largest observed increases in tether-to-fusion time (*Figure 8A*).

As Mso1 forms a tight complex with Sec1 to aid Sec1 in SNARE assembly, we considered two models for how the loss of Mso1 could affect Sec1 and, in turn, vesicle fusion (*Figure 8B*). In one scenario, loss of Mso1 may delay the arrival of Sec1 to the tethered vesicle but, once recruited, Sec1 may take the same amount of time to perform its essential functions. In the second scenario, if Mso1 participates in Sec1's essential function, its absence could cause Sec1 to be prolonged at the tethered vesicle, thereby delaying fusion. A defining feature of these two models is the longevity of Sec1 puncta on the plasma membrane. When we examined the longevity of Sec1-mNG in an *mso1* null, we saw no such change in the time from Sec1 punctum appearance to disappearance (*Figure 8C*). Nor were there any apparent changes in the number of Sec1 molecules per punctum in the *mso1* null strain (*Figure 8D*). We did, however, find that broad plasma membrane localization of Sec1 (outside of discrete puncta) was diminished in the *mso1Δ* strain, especially for cells with midsized buds, in which vesicle tethering times were normally measured (*Figure 8E*).

These data suggest that Mso1s primary function may be to increase the local concentration of Sec1 on the plasma membrane near sites of polarized growth, hastening its ultimate recruitment to a tethered vesicle thereby promoting efficient secretory vesicle fusion. This is further supported by our new observations concerning Sec1 membrane recruitment. Sec1 has a positively charged C-terminus, and Mso1 has an amphipathic N-terminus (*Figure 8F*), either of which when visualized independently localize broadly to the plasma membrane (*Figure 8G*). Individual loss of either of these regions from Sec1 or Mso1, can be tolerated, but simultaneous loss of these two membrane-associating regions is lethal (*Figure 8H* and *Figure 8—figure supplement 1*). It is interesting to note that the extended Sec1 tail is unique to fungi and all Sec1 C-termini appear to contain this highly positively charged sequence, although the length of the preceding linker region varies (*Figure 8—figure supplement 2A*). Mso1 is also unique to fungi, specifically ascomycetes, but is considerably more variable between species, with the Sec1-binding domain being the principal highly-conserved motif within the protein sequence (*Figure 8—figure supplement 3*). This poses interesting questions concerning the evolutionary divergence of exocytic mechanisms and regulators; why the Sec1 tail is found in non-ascomycetes, while Mso1 appears limited to this phylum is not immediately apparent.

## Discussion

Since many of the components of the yeast secretory pathway were first described in the 1980's, attempts to further characterize the proteins have relied primarily on genetic and biochemical dissection of interactions to order the events leading to vesicle fusion at the plasma membrane. The dense and fast-growing nature of the yeast bud, however, has largely prevented direct visualization of secretory components of single exocytic events. Given the relatively large number (~15–30; *Govindan et al., 1995*; *Mulholland et al., 1994*) of secretory vesicles which are present within the bud at any moment to maintain growth and counteract the internalization of membrane from endocytosis (which also primarily occurs within the bud), resolving individual vesicles is challenging.

At any point, there may be near two dozen secretory vesicles in a growing bud and when this is considered alongside the speed at which vesicles are capable of moving via Myo2 (*Schott et al., 2002*)—an average of 3μ/s—and the diffraction-limited resolution of conventional microscopy, sufficiently fast capture time can be the difference between observing stationary tethered vesicles and a blurred mass of signal. For the best spatiotemporal resolution, we opted to use high-speed spinning disk-confocal microscopy with short exposures and high excitation energies to image exocytic events over short time windows (30 s-1m). Imaging only the half of the bud closest to the coverslip also simplified the task of visualization. Analysis of all single-fluorophore microscopy was performed by viewing videos in 3D, as opposed to simple max or sum projections and this technique was invaluable for the ability to track individual vesicles through space, define tethering events with high confidence, and rule out abortive tethering events. Super-resolution microscopy techniques, while desirable for the increased spatial resolution, are yet unable to capture such events with sufficient temporal resolution.

When possible and when they did not appear to compromise function or localization, we used the brightest yeast codon-optimized and monomeric fluorescent proteins currently available: mNeonGreen and mScarlet (*Bindels et al., 2017*; *Lambert, 2019*; *Shaner et al., 2013*). Furthermore, all proteins were tagged directly in the genome under the expression of their own promoters, despite

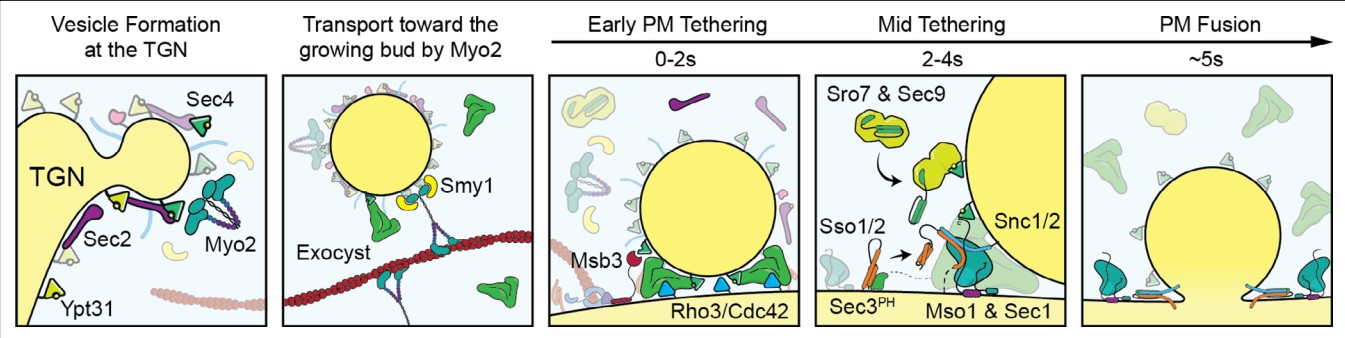

**Figure 9.** Visual timeline of secretion from Golgi to plasma-membrane fusion. Active Ypt31/32 on Trans-Golgi Network (TGN) membranes recruits the secretory Rab-GEF Sec2, which thereby recruits Sec4 to the TGN ahead of vesicle formation. Though vesiculation of the Golgi occurs by as-yet-unknown mechanisms, newly activated Sec4 recruits the myosin-V motor protein Myo2 to the Golgi around the time of separation from TGN compartments and the Myo2-vesicle interaction is enhanced by the kinesin-related protein Smy1. During transport to the plasma membrane of the bud the complete exocyst complex, including Sec3, is recruited to vesicles through interactions with multiple proteins including Sec4, Myo2, and Sec2. Upon reaching the plasma membrane, Rho-proteins on the membrane (Rho3, but likely Cdc42 as well) concentrate slightly at the vesicle interface facilitating the activation of the exocyst, thereby aiding tethering. The initial plasma-membrane recognition and selection of tethering sites likely involves many other inputs, including but not limited to the PIP composition of the membrane, position of actin-cable termini, and scaffolding proteins like Boi1/2. Shortly after tethering, Sec2, Smy1, and Myo2 dissociate from the tethered vesicle and Msb3 begins inactivating portions of the vesicle-bound Sec4 population. Sro7 (and Sec9) are then recruited to the vesicle-PM interface through active Sec4 as well as multivalent interactions with the exocyst. SNARE assembly at the interface is subsequently aided by the exocyst—including Sec6 and Sec8, as well as Sec3, which is partially responsible for activating Sso1/2—and, finally, Mso1. Shortly before fusion, Mso1 and Sec1, which are enriched on the plasma membrane of the bud by their respective membrane-binding domains, begin accumulating at the vesicle interface and rapidly dissipates around the moment of exocyst complex disassociation and vesicle fusion. After membrane fusion, any remaining active Sec4 is returned to an inactive state by Msb3/4 to facilitate removal from the plasma membrane.

this severely restricting the number of molecules available to image. In part due to this imposed limitation, correlative imaging of any combination of two proteins with separate fluorescent tags became an even more complicated task. Under ideal conditions (excitation laser $\lambda$, emission filter $\lambda$, equal camera sensitivity), mScarlet is still roughly 75% as bright as mNeonGreen and, in our hands, mScarlet fusion proteins were generally less well-behaved. Tagged mScarlet-Sec4 did not even appear as bright as GFP-Sec4, perhaps owing to mScarlet's much longer fluorescence maturation time (*Lambert, 2019*). For these reasons, very few protein pairs were possible to image while maintaining reasonable temporal resolution.

During the formation of secretory vesicles at Ypt31-marked compartments, the secretory Rab-GEF Sec2, precedes arrival of its cognate Rab, Sec4, by about a second (*Figure 9*). The direct recruitment of Sec4 to the Ypt31 compartment is apparently distinct from mechanisms described in *Aspergillus nidulans*, where recruitment of these Rabs is spatially separated. In this filamentous fungus, RabE, the homolog of Yp31, is transiently recruited to the trans-Golgi Network where it picks up actin- and microtubule-based molecular motors (*Pantazopoulou et al., 2014*). This compartment is then moved apically to the vesicle supply center (or Spitzenkörper), where RabD, the homolog of Sec4, is then recruited (*Pinar and Peñalva, 2021*). Following arrival of newly activated Sec4, the myosin-V motor protein, Myo2, is recruited to the membrane by Sec4, though its rapid arrival is likely supported by active Ypt31/32 (*Lipatova et al., 2008*). Additionally, while Myo2 appears capable of localizing to these compartments ahead of vesicle separation from the TGN, the force generated by Myo2 transport does not appear to be necessary for vesicle formation, as this process continues in the absence of actin cables. We have also found that the entire exocyst complex (including Sec3) localizes to secretory vesicles during transport to the bud and remains associated with the vesicle throughout tethering, only disassociating near the moment of fusion. While this does not necessarily rule out the presence of a population of Sec3 that localizes to the plasma membrane independently, the vesicular localization and the FLIP data appear to support a model where the yeast exocyst is an obligate hetero-octamer (*Heider et al., 2016*).

In this study, secretory vesicles were found to tether for about five seconds before fusion with the plasma membrane; a far shorter time than previously believed. Prior work had indicated a 'long' tether-to-fusion time for vesicles on the order of 15–18 s (*Alfaro et al., 2011*; *Donovan and Bretscher, 2015a*). However, it now seems far more likely that such results are artefacts of slow imaging

speed and the presence of vesicle tethering hot-spots. These hot-spots are likely biologically favorable for the maintenance of efficient and productive vesicle fusion. Sequential and/or simultaneous tethering events would logically permit utilization of the same pool of cofactors responsible for the initiation of tethering and downstream events preceding fusion.

Sec2, Smy1, and Myo2 all dissociate from secretory vesicles shortly after tethering, though release occurs at different rates and is not a concerted event. In future studies, it will be interesting to see how displacement of Sec2 affects release of other components and timing of overall vesicle tethering as a whole. Since deletion of the Sec4 GAP Msb3 elongates the vesicle residence time of every component measured except Sec2, it appears that Sec2 release is controlled by a Sec4-independent process, perhaps dephosphorylation by an as yet unknown bud-resident phosphatase, a regulatory mode previously suggested (*Stalder et al., 2013*; *Stalder and Novick, 2016*). Further, since GAP function necessarily requires binding to the same face of the Rab recognized by most effectors (*Pan et al., 2006*), a direct competition is implied since Msb3 cannot simply "kick off" Myo2 and others bound to Sec4 at the plasma membrane by stimulating Sec4-GTP hydrolysis allosterically. Thus, exactly how Msb3 function and Sec4:GTP state are coupled to tethering time needs to be explored further.

Loss of Msb3 elongates the tethering time by acting on tethered vesicles prior to fusion. Although loss of Msb3 also results in accumulation of Sec4 in the plasma membrane, we have shown that this is not the cause of elongated tethering. If we take as axiom that Sec4:GDP readily binds new GTP without the aid of its GEF Sec2 (*Kabcenell et al., 1990*; *Rinaldi et al., 2015*; *Walch-Solimena et al., 1997*), then some portion of Sec4 delivered to the plasma membrane through fusion is likely to be Sec4:GTP, preventing extraction by GDI (Guanine-nucleotide Dissociation Inhibitor). Therefore, Msb3 acts on Sec4 'twice', once on the vesicle pre-fusion and once on the plasma membrane post-fusion. We believe this to be the simplest explanation of the observed data. Since efficient hydrolysis of Sec4:GTP is not strictly essential for exocytosis (*Sec4*$^{Q79L}$ dramatically reduces GAP-stimulated GTP hydrolysis and GEF GDP release, yet haploids with this allele are viable), one interpretation is that the action of Msb3 on vesicle-bound Sec4 aids in the facilitated release or swapping of Sec4 effectors. The effectors likely to be bound by Sec4 after vesicle arrival at the plasma membrane could include Sro7 and the Sec1 accessory Mso1 (*Rossi et al., 2018*; *Weber-Boyvat et al., 2011*).

The unexpected difference between the effects of two conditions which, in principle, both increase relative Sec4:GTP abundance (Sec2 overexpression and loss of Msb3) highlights the spatiotemporal regulation and robustness of secretion. Whereas deletion of plasma membrane localized Msb3 shifts the Sec4 equilibrium towards the GTP-bound state in the context of tethered vesicles, overexpression of Sec2 shifts the GTP equilibrium in the context of vesicle formation and transport. The early presence of Sec2 and the maintenance of Sec4:GTP during transportation to the bud tip directly aids in the recruitment of crucial effectors like Myo2 and the exocyst. Such Sec4:GTP maintenance is unfavorable in the context of a tethered vesicle, perhaps due to the exchange of Sec4 effectors near the plasma membrane. Correspondingly, when we look at the timing of individual components in the context of an *msb3ΔΔ*, we can see that the rapid release of Sec2 from secretory vesicles after reaching the bud tip is uncoupled from the downstream events preceding vesicle fusion, as it is the only component for which there is no significant change.

By combining the rapid three-dimensional capture and analysis of individual components with fast 2D dual-color microscopy we were next able to align several events which occur before fusion vesicle fusion to generate a timeline from tethering to fusion. Strains with multiple subunits of the exocyst tagged with mScarlet proved to be the most useful tool for alignment of these exocytic events. This set of three tags (comprised of fusions with Sec10, Sec15, and Exo84) localizes well and appeared to behave fine when combined with most other fusions, only moderately affecting timing of various events.

While Rho1 and Cdc42 are capable of interacting with the exocyst through the N-terminal PH-like domain of Sec3, Rho3 is of particular interest due to its ability to bind the exocyst through Exo70, a subunit which appears to be involved in the activation of the exocyst (*Robinson et al., 1999*; *Rossi et al., 2020*; *Wu et al., 2010*). Previous studies have shown that gain of function mutants in Exo70 are capable of suppressing loss of Rho3 and that the same gain of function mutations cause the exocyst to shift into a partially open 'active' conformation which exposes new binding sites. We previously showed that Rho3 primarily localizes to the plasma membrane, and not on internal vesicles, with concentration on the membrane increasing toward the bud tip and transient discrete puncta

of yet higher concentration (*Gingras et al., 2020*). Imaging of Exocyst-3x-mScarlet with endogenous expression of this previously developed Rho3-imNG showed that Rho3 initially concentrates in puncta at vesicles shortly following their arrival to the plasma membrane. Together, a plausible model suggests that diffusing Rho3 on the plasma membrane is slowed down through interaction with Exo70, resulting in an apparent local accumulation, after which the interaction induces a conformational change in the exocyst, potentially to facilitate the exocysts binding of SNARE complexes and other exocytic proteins.

Subsequent to this, Sro7, a homolog of the *lethal giant larvae* and tomosyn polarity protein, is recruited to the exocytic site. As Sro7 primarily interacts with the exocyst through the exposed and labile N-terminus of Exo84 (*Mei et al., 2018*; *Rossi et al., 2015*; *Zhang et al., 2005*), it seems unlikely that the aforementioned conformational change in the exocyst is responsible for triggering binding of Sro7. Rather, it seems plausible that loss of Sec2 and Myo2 from the vesicle frees a population of active vesicular Sec4 which then enhances the recruitment of Sro7 as a new effector, in concert with Exo84; this would be consistent with observed timing of Sro7 accumulation (*Rossi et al., 2018*; *Watson et al., 2015*). How Msb3 plays a role in this transition is unclear, however, a direct role seems likely based on the increase in Sro7 punctum longevity observed in *msb3* null cells. Interestingly, despite Sro7 localizing to sites of exocytic vesicles mid-tether, some 20% of Sro7-mNG puncta appeared to lack clear exocyst colocalization. Sro7 has been claimed to function in parallel to the exocyst, and overexpression can bypass loss of certain exocyst components, so it's possible that these puncta represent a separate class of vesicle of uncertain identity which remains more dependent on Sro7 function (*Grosshans et al., 2006*; *Lehman et al., 1999*).

While tethering, as facilitated by the exocyst complex, is the first and more reversible step towards vesicle fusion, docking, which is thought to be facilitated by SNARE-assembly, is likely more stable. SNARE proteins are not capable of stimulating membrane fusion on their own in vivo, instead requiring the aid of additional factors called Sec1-Munc18 (SM) proteins (*Baker and Hughson, 2016*; *Hong and Lev, 2014*). Notably, the exocyst itself plays several important roles in SNARE assembly, including initial Sec3-stimulated disinhibition of Sso1/2 (*Yue et al., 2017*) and direct SNARE complex assembly independent of Sso1/2 opening (*Lee et al., 2022*), although the myriad interactions responsible for this assembly have been better discussed in recent publications and preprints (*Lee et al., 2022*; *Rossi et al., 2020*).

Though not directly imaged, the SNARE Sec9, Sro7, and Sec4:GTP have been shown to form a ternary complex, so the arrival of Sro7 at the tethered vesicle likely indicates the arrival of Sec9 (*Grosshans et al., 2006*). The local recruitment of Sec9 thereby increases the likelihood of forming the binary and ternary SNARE-pin intermediates capable of binding both Sec1—which has essentially no interaction with SNARE monomers but is essential for exocytosis and viability—and Sec6 of the exocyst (*Carr et al., 1999*; *Dubuke et al., 2015*; *Togneri et al., 2006*). Crystallographic studies of the vacuolar SM protein Vps33 suggest that SM proteins function as templates of SNARE assembly (*Baker et al., 2015*; *Baker et al., 2013*). Earlier studies, however, have shown that Sec1 has distinct functions both before and after 'docking', where docking is defined as SNARE-pin assembly (*Hashizume et al., 2009*). Additionally, early studies of the neuronal Sec1 in rats (Munc18) and *Drosophila* (Rop) suggested that SM proteins may have a negative-regulatory role in exocytic regulation (*Halachmi and Lev, 1996*; *Schulze et al., 1994*; *Zhang et al., 2000*).

This study contains the first direct in vivo visualization of SM proteins with sufficient spatiotemporal resolution to identify localization to single exocytic vesicles. The data presented here on the timing and dynamics of Sec1 function in yeast, support a model where SM proteins accumulate at pre-fusion membranes helping to facilitate SNARE assembly, holding in place briefly before concerted release, and thereby, membrane fusion. Since Sec1 has such a low affinity for individual SNARE motifs, direct 'templating' through Sec1 seems unlikely, though it is possible it plays some role in directing alignment of the zero-layer SNARE residues. It is far easier to imagine why it would be beneficial for Sec1 to remain associated with SNARE-pins preassembled with the aid of the exocyst complex (*Yue et al., 2017*), as the presence of the SM protein should prevent the unwanted disassembly of otherwise productive trans-SNARE complexes by NSF and α-SNAP, Sec18 and Sec17, respectively, in yeast (*Jun and Wickner, 2019*; *Song et al., 2017*). Sec1 is aided in initial plasma membrane localization by the small ascomycete-specific peripheral plasma membrane protein, Mso1, and that this initial membrane recruitment represents a shared essential function of Mso1 and the (also fungal-only) Sec1 C-terminus

(*Figure 9*). In higher eukaryotes, this initial localization is instead accomplished through a direct interaction between the exocytic SM protein and an N-terminal peptide on the Syntaxin-homologs (which is not found in the yeast Sso1/2 syntaxins)(*Hu et al., 2007*; *Rathore et al., 2010*). Mso1 has also been previously reported to play a role in the assembly of SNARE complexes (*Castillo-Flores et al., 2005*; *Weber-Boyvat et al., 2011*).

Our timeline and relative abundance of participating components involved in the biogenesis of secretory vesicles at the Golgi and their exocytosis at the plasma membrane raises many questions. Notably, the exact mechanism of secretory vesicle biogenesis is still shrouded in mystery. Previous searches for proteins responsible for secretory vesicle biogenesis have relied on the assumption that if secretion itself is essential, the formation of secretory vesicles must also be essential. Additionally, previous studies have suggested the existence of multiple secretory pathways, a model which is supported by some observations in this study. Although we did not uncover any clear evidence of differentially regulated tethering and fusion events, future studies (perhaps utilizing cargo-specific markers) may still find subtle but physiologically significant differences in the regulation of vesicle subpopulations. The solutions of these mysteries, as well as mechanistic and structural studies on how the conformational transition of the exocyst facilitates the shift from tethering to docking in fusion are all questions that can build on the framework presented here.

## Materials and methods

### Yeast strains, growth, and transformation

Yeast strains used in this study are listed in *Supplementary file 1*. Standard media and techniques for yeast growing and transformation were used. Gene deletion and C-terminal chromosomal tagging was performed using common PCR-mediated techniques (*Longtine et al., 1998*). See *Supplementary file 1* for oligos used for amplification of integration cassettes. Tetrad dissections were performed using an MSM-400 dissection scope (Singer Instruments, Somerset, United Kingdom) with 25 µm needle following a one-week incubation at 26 °C in standard sporulation media (1% yeast extract, 1% potassium acetate, and 0.05% glucose). Strains are available upon request to the corresponding authors.

### DNA constructs

Plasmids used in this study are listed in *Supplementary file 1*. The integrating plasmid pRS306-GFP-Sec4 used to tag Sec4 has been previously described (*Donovan and Bretscher, 2012*). The Rho3-imNG constructs as well as the yeast codon optimized mScarlet were described previously (*Gingras et al., 2020*). N-terminal tags of Ypt31 and Ypt32 were amplified from plasmids containing a selectable marker, the relevant promoter, and associated tag via oligos containing additional genomic homology and then integrated via transformation. For construction of pRS415-pCYC100-Tomato-mouseCC-Myo2(Cargo Binding Domain)-tCYC, first codon-optimized sequence (See *Supplementary file 1*) coding for 156 residues of the coiled-coil region of Mouse Myosin 5b was synthesized by IDT (Coralville, Iowa). This fragment was then amplified with oligos (RMG744/745) containing homology to the sequence flanking the endogenous Myo2 coiled-coil and recombined into a plasmid containing the Myo2 locus. From this intermediate plasmid, the coiled-coil through the Myo2 cargo-binding domain was amplified (oligos RMG761/762), one copy of Tomato was amplified (RMG759/760), and both were Gibson cloned into a linearized pRS415-pCYC100-(MCS)-tCYC. Tomato was later replaced with mNG via recombination (oligos RMG870/871) and the whole cassette was restriction cloned into pRS303 for genomic insertion into the *His3* locus. All plasmid sequences are available upon request to the corresponding authors.

### Construction of Exocyst-3x-mNG/mScarlet strains

During the initial planning of the Exocyst-3x-tag strains, Sec10 and Sec15 were chosen first. For issues of sterics, we initially excluded Sec5 and Exo70 from a multi-tag complex. Sec6 was cautiously excluded due to important interactions with SNAREs and Sec1 which occur on the C-terminal portion of the protein (the exact residues required for these interactions is unknown). Exo84 became our third tagged protein, although it is possible that Sec3 or Sec8 could have worked equally as good while also providing a marker for the other exocyst subassembly. As the goal of these strains was to

provide an additional marker suitable two-color imaging, we did not feel this was necessary to take into consideration.

Haploid cells containing C-terminal endogenously tagged *Exo84-mNG::Ura3* and *Sec15-mNG::Ura3* were mated and sporulated to obtain a strain with both tags. The *Exo84-mNG::Ura3 Sec15-mNG::Ura3* haploid was then mated with *Sec10-mNG::Ura3* and sporulated to obtain a strain with all three tags. A guide RNA was created to target the *mNG* sequence 20 bps from the linker region with BplI cut sites at both 5' and 3' ends. After ligating the gDNA oligos, the guide was digested with BplI and cloned into the CRISPR-Cas9 vector bRA90 which expressed this gRNA and the CRISPR machinery (*Anand et al., 2017*). A separate repair *mScarlet::NatMX* was made with 40bps homologous sequences to the linker region and the beginning of *mNG* and the last 40bps of homology to *Ura3*. The bRA90 vector (200 ng) and the repair (1 ug) was transformed into the *Exo84-mNG::Ura3 Sec15-mNG::Ura3 Sec10-mNG::Ura3* haploid cells. Cells were plated on NatMX -LEU MSG plates and grown at 26 °C for 1 week. Transformants were screened for red fluorescence and absence of green and positive clones were selected for further experiments. The bRA90 plasmid was eliminated from the cells by repeatedly plating them on NatMX plates until the LEU plasmid was lost. See *Supplementary file 1* for oligos used for gRNA and template creation.

## Microscopy techniques

Most micrographs in this study were acquired on a CSU-X spinning-disk confocal microscopy system (Yokogawa, Tokyo, Japan; 3i Intelligent Imaging Innovations, Denver, Colorado) with a DMI6000B microscope (Leica Microsystems, Wetzlar, Germany), 100×1.45 NA objective (Leica), and an Evolve 512Delta EM-CCD (Teledyne Photometrics, Tucson, Arizona) with a 2×magnifying lens (Yokogawa) resulting in 0.084 µm/pixel. A few images (Supplement 2.1 and *Figure 3B*) were captured instead on a Flash 4.0v2 CMOS (Hamamatsu, Hamamatsu City, Japan) with a final resolution of 0.065 µm/pixel. For live-cell imaging, cells in mid–log phase were adhered to a glass-bottomed dish (CellVis, Mountain View, California) coated with concanavalin A (EY laboratories, San Mateo, California) and washed with the respective dropout or complete cell medium. Unless otherwise stated, imaging of single-channel fluorescence was performed via SlideBook 6.0 'Rapid 4D' capture, with sustained laser intensity, constant piezo stage movement, and video frames streamed to RAM, with 25–50ms exposure per plane. As these videos were captured with the bottom plane of the volume coincident with the bottom of the cell, the result was a 6-plane, 1.5 µm volume of the bud, captured with 175–333ms per time-point. For two-channel microscopy, single planes focused on the bottom or center of the cell were captured in rapid alternation with 75–100ms exposure per channel. Simultaneous capture of two colors was not possible on our system, as we lack a beam splitter. These alternating images were later aligned for display in figures.

The 'Rapid 4D' live cell imaging, described above, was performed with the EM-CCD, despite the need to use a 2 x magnifier to compensate for the large pixel size of the detector. The EM-CCD used provides a global shutter and thus simultaneous capture of the field to be imaged. With a continuously moving stage, this means each subsequent image on this camera represents a discrete position along the vertical axis which could subsequently be assembled into a 3D volume with the knowledge of stage position at each timepoint (recorded as piezo voltage during capture). As our CMOS utilizes a rolling shutter (with pixels first being read linearly from the center of the image outward) an image taken with continuous stage movement would result in single frames containing information from several vertical positions, significantly complicating volumetric reconstruction.

## Video and image analysis

Molecule counting was performed by comparison to Cse4-mNG puncta intensity at anaphase, as described previously (*Donovan and Bretscher, 2012*). All timings were performed on blinded image data with file names randomized with a freely available Perl script (*Salter, 2016*). All blinded tethering and longevity captures were analyzed using 3D projection in SlideBook 6.0 when possible. About 50 events (from videos captured on at least three separate days) were quantified per component or genotype. Generally, videos were played in reverse to identify events that concluded within the capture time then, once single events were confidently identified (based on lack of nearby fluorescence or component movement), the tethering start point was determined. Tethering start was defined as the first frame in which a GFP-Sec4 vesicle (or another component residing on a vesicle) came to a full

stop at a location not more than one apparent vesicle diameter from where it eventually fused (or signal disappeared). Fusion (or disappearance) was defined as the last frame in which signal could be positively identified at the tethering location.

Plots of signal over time for Sro7 and Sec1 (*Figure 5D and F*) were generated by extracting punctum intensity of events from sum projections of the bottom half of buds in FIJI (captured as in *Figure 2B*, videos blinded). After measuring the intensity over time for many puncta, several events were grouped with apparent lengths of one standard deviation of the mean longevity (of the events measured this way). These intensity profiles were then aligned manually by apparent moment of peak intensity, normalized to peak, and averaged in Prism 9 (GraphPad Software, San Diego, California). The dual channel intensity plots in *Figure 6* were similarly generated, except the source videos were single plane captures (described above) and alignment of similarly timed events was performed based on either the apparent start of tethering (for combinations with Sec4 or Exocyst) or peak Sec1 intensity.

## Latrunculin B (LatB) experiment

Experiments assessing the effects of actin cable loss on tethering were performed with addition of latrunculin directly to a dish containing a small volume of cells on the microscope stage to permit rapid imaging of subsequent timepoints. Briefly, 50 µL of LatB in ethanol (Cayman Chemical, Ann Arbor, Michigan) was evaporated under nitrogen and resuspended in an equivalent volume of synthetic complete media. To facilitate rapid diffusion, 8 µL of this resuspension was added directly to 92 µL of cells attached to a dish as above, for a final concentration of 100 µM LatB. Under these conditions most actin cables are lost within the first minute.

## Fluorescence loss in photobleaching (FLIP) experiment

For the FLIP experiment with Sec3 and Sec5, since individual vesicles did not need to be distinguishable GFP-fusions were chosen. FLIPs were performed essentially as done in prior studies (*Donovan and Bretscher, 2015b*). Briefly, using a 3i Vector photobleaching system, the back half of the mother cell was photobleached once every 2 s with simultaneous monitoring of bud intensity. Pre-photobleach intensity of signal within the bud was set to 1.0, with background fluorescence subtraction, and further photobleaching normalization to a nearby non-targeted cell. These measurements were later averaged and depicted ± SD.

## Statistics and presentation

Graphs and statistical analyses were generated in Prism 9. Except where stated, non-parametric tests were used due to non-normality of some data sets. Paired t-tests were used to compare the relative Ypt32/Ypt31 TGN intensities between mother cells and buds (*Figure 1—figure supplement 4C, D*). For all single component timings (compiled in *Figure 5—figure supplement 2*), general differences were first assessed via Kruskal-Wallace non-parametric ANOVA, followed by the select Dunn's corrected post-hoc tests shown. Similarly, vesicle tether-to-fusion timings in deletions and overexpressions were grouped (as shown in *Figure 7—figure supplement 1*), with deletions being compared to the wildtype vesicle tether-to-fusion times first shown in *Figure 2C*, and each overexpression being compared to the relevant 2 µm empty vector control. In this case, corrected post-hoc tests were performed between the relevant control and each experimental condition only (not between any independent overexpression or deletion conditions). Due to non-normality, comparisons for individual component timings in wildtype versus in *msb3Δ* cells were performed via Mann-Whitney test (*Figure 7D*). Exact sample sizes for all measurements not otherwise explicitly stated in figures are shown in *Supplementary file 1e*, values measured are shown in *Supplementary file 1f*. Fungal Sec1 and Mso1 sequences were gathered from UniProt (*Bateman et al., 2021*), aligned via Clustal Omega (*Sievers et al., 2011*), and colorized with JalviewJS hosted on MyHits (*Pagni et al., 2007*; *Waterhouse et al., 2009*). Figures were assembled in Illustrator (Adobe).

## Acknowledgements

This work was funded by NIH grants 5RO1GM039066 and 5R35GM131751.

# Additional information

## Funding

| Funder | Grant reference number | Author |
|---|---|---|
| National Institute of General Medical Sciences | 5RO1GM039066 | Anthony Bretscher |
| National Institute of General Medical Sciences | 5R35GM131751 | Anthony Bretscher |

The funders had no role in study design, data collection and interpretation, or the decision to submit the work for publication.

## Author contributions

Robert M Gingras, Conceptualization, Formal analysis, Supervision, Validation, Investigation, Visualization, Methodology, Writing - original draft, Writing - review and editing; Abigail M Sulpizio, Investigation, Methodology, Writing - review and editing; Joelle Park, Formal analysis, Investigation, Writing - review and editing; Anthony Bretscher, Conceptualization, Supervision, Funding acquisition, Validation, Writing - original draft, Project administration, Writing - review and editing

## Author ORCIDs

Robert M Gingras ![ORCID] http://orcid.org/0000-0002-7377-0845
Joelle Park ![ORCID] http://orcid.org/0000-0003-0221-6967
Anthony Bretscher ![ORCID] http://orcid.org/0000-0002-1122-8970

## Decision letter and Author response

Decision letter https://doi.org/10.7554/eLife.78750.sa1
Author response https://doi.org/10.7554/eLife.78750.sa2

# Additional files

## Supplementary files

• Supplementary file 1. All supplementary files described in methods. (1a) Yeast strains, (1b) plasmids, and (1c) oligos used in this study. (1d) Mouse Myosin-Vb sequence used for generating the fluorescent Myo2 markers used. (1e) All sample sizes for statistical tests. (1f) All raw tethering and component lifetime data.

• MDAR checklist

## Data availability

Details of each yeast strain used is provided in Supplementary File 1a. All plasmids used are listed in Supplementary File 1b. All DNA oligos used are listed in Supplementary File 1c. The unique Mouse Myosin-Vb sequence used for the Myo2 marker is provided in Supplementary File 1d. All sample sizes for data and statistical tests shown in text are provided in Supplementary File 1e. All raw tethering data and component lifetime timing data, as well as calculated mean and medians are provided in Supplementary File 1f.

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
