## [Editor Report]

This paper describes a tour de force characterization of the timeline and molecular events during the delivery and fusion of yeast secretory vesicles. It establishes a standard for further investigation of this important process.

---

## [Decision Letter]

**Decision letter after peer review:**

Thank you for submitting your article "High-resolution secretory timeline from vesicle formation at the Golgi to fusion at the plasma membrane in *S. cerevisiae*" for consideration by *eLife*. Your article has been reviewed by 3 peer reviewers, including Benjamin S Glick as the Reviewing Editor and Reviewer #1, and the evaluation has been overseen by Vivek Malhotra as the Senior Editor.

Essential revisions:

Three reviewers have evaluated your manuscript, and all of us are enthusiastic about the work. It is a substantial advance both technically and conceptually.

As you will see from the reviews, there are issues to be clarified as well as suggestions for improving the presentation. Please consider these comments carefully and revise the manuscript according to your best judgment. The changes will be primarily at the level of the text and figures, although some minor additional experimentation might be merited. The main points are as follows.

– Nonspecialists will find it challenging to keep track of the various components and their roles in the pathway. Additional background would be helpful to orient readers. You might consider providing a table that lists the components with brief summaries of their functions. In addition, it will help to define terms explicitly and to acknowledge your assumptions. An example is to explain how you are assessing "tethering" and "fusion" experimentally-e.g., if fusion is defined as loss of Sec4-GFP signal at the plasma membrane, that's an important assumption that should be stated and justified.

– On a related note, the manuscript would benefit from a mechanistic diagram that pulls together the new findings into a model. Some speculation is fine here. The idea is to emphasize how the results deepen our understanding of the journey experienced by a secretory vesicle.

1) Given the emphasis in this work on imaging technology, it would be useful to provide full explanations about how the microscopy system was configured and why particular choices were made.

2) Studies of related phenomena in Aspergillus should be cited and considered. It's likely that many basic aspects of secretion are conserved between yeast and Aspergillus, but the Aspergillus system has unique properties that can illuminate features of the pathway.

3) Further discussion of Ypt31/32 would be helpful. The reviewers were surprised by the conclusion that Ypt31/32 remains on secretory vesicles. Perhaps the GFP tag partially compromises function and inhibits GAP activity? I note that GFP tags have been reported to compromise Ypt1 function, and we see something similar for Ypt6. Another point is that the Ypt31/Ypt32 pair arose from a relatively recent whole-genome duplication, so it's unlikely that the two paralogs have substantially different functions or that such differences are relevant beyond yeast. This concept is worth mentioning.

4) It would be helpful to distinguish more clearly between firm interpretations and speculations, especially in cases where you are pushing the technology to the limit and making inferences about what you are seeing.

5) Some of the conclusions echo prior work from other investigators, and those earlier contributions should be better acknowledged. A particular example is the idea that Sec3 is not actually a landmark component (see Roumanie et al., 2005).

*Reviewer #1 (Recommendations for the authors):*

"TRAPPII" has two P's.

Regarding the subtle differences between Ypt31 and Ypt32: is it possible that Ypt32 associates with secretory vesicles more efficiently than Ypt31, with the result that the local concentration of Ypt32 in the bud is higher than that of Ypt31?

Line 188: "Figure 5D" should presumably read "Figure 2D".

I don't quite understand Figure 2E. Isn't Sec4 present on the vesicle before tethering? If so, why is the Sec4 intensity very low just prior to tethering? Maybe the zero on the horizontal axis is in the wrong place?

In Figures 4, 7, and 8, I believe the convention for yeast is to capitalize all letters of a wild-type gene name rather than just the first letter.

*Reviewer #2 (Recommendations for the authors):*

Additional discussions of the generality of these findings, e.g. will this information be similar for non-yeast eukaryotes, would be helpful.

Could the Ypt31 on the plasma membrane (or near the plasma membrane) represent recycling vesicles?

For the Exocyst-3x-tag construct, why were the 3 subunits all chosen to be from the same subcomplex rather than some from each subcomplex? Presumably, the exocyst is octameric, but this construct would only report on half of it.

The Rho3 timing vs. exocyst seems to be too noisy to be able to clearly conclude that Rho3 peaks ~1 sec after tethering starts.

The genetic experiments (heterozygous and homozygous deletions, and overexpression) are interesting, but difficult to make unambiguous conclusions from, especially if the levels of the proteins in the heterozygous cells are unknown, and relative overexpression levels vs endogenous proteins are also unknown.

In line 482, the reference for the "new observations" paper is missing.

The overall writing could be tightened up, and several typos are scattered throughout.

*Reviewer #3 (Recommendations for the authors):*

– Ypt31 but not Sec7 remaining upon vesiculation? This does not match with the fact that Sec7 is a Ypt31 effector (Fromme, Dev Cell)

– Figure 1B tracking of SEC2 not convincing, perhaps related to timestamp differences between dual-color images. Figure 6C tracking of Rho3 is not convincing either. Why not use a beam splitter?

– Please clarify further if there are separated roles for Ypt31 and Ypt32.

– Please integrate into the timeline the fact that Myo V is a direct effector of Ypt31/32 (Lipatova and Segev, 2008) lines 99-100.

– Please explain why a 'sizeable population of Ypt31 remains on the membrane' (Figure 1A).

– Line 128 Hugh Pelham denoted these compartments as the post-Golgi endosome, fed by membranes derived both from the Golgi and the plasma membrane. They might be equivalent to rapid recycling endosomes of metazoans.

– It is crucial to establish whether detachment of Sec2 compartments occurs before motor engagement, thus this part should be further expanded, or removed if inconclusive

– 211 alone alone repetition.

– 327 why very few? compared to what?

– 338 perturbation (typo); (does it mean synthetic negative interaction)? why 'apparent'?

– Figure 6S2, add linescan.

– how do you power vesiculation?

– 420- lack of effect of Sec2 overexpression: GFP-Sec4 insensitive to nt exchange?

– Figure 7S1 B MyoE facilitates (shortens) the tether to fusion transit; how do you envisage it mechanistically?

– Please provide a readable file 8S2. The wording is confusing here, yeast vs. fungi? Mso1 appears to be absent from filamentous fungi, you may cite FungiDB (https://fungidb.org/fungidb/app/record/gene/YNR049C#Orthologs).

– I missed the evidence that Mso1 is a Sec4 effector – Figure 8I?

– 501… large number of secretory vesicles per unit time: would authors endeavor to provide an estimation?

– 530: a beam splitter may have been useful.

– 534 Ypt31 marked secretory compartments might mislead readers.

– 578 Has it been formally established that Q79L prevents GTP hydrolysis? (see Barr, 2014, Diversity and plasticity in Rab GTPase nucleotide release mechanism has consequences for Rab activation and inactivation)

---

## [Author Response]

Essential revisions:Three reviewers have evaluated your manuscript, and all of us are enthusiastic about the work. It is a substantial advance both technically and conceptually.As you will see from the reviews, there are issues to be clarified as well as suggestions for improving the presentation. Please consider these comments carefully and revise the manuscript according to your best judgment. The changes will be primarily at the level of the text and figures, although some minor additional experimentation might be merited. The main points are as follows.– Nonspecialists will find it challenging to keep track of the various components and their roles in the pathway. Additional background would be helpful to orient readers. You might consider providing a table that lists the components with brief summaries of their functions. In addition, it will help to define terms explicitly and to acknowledge your assumptions. An example is to explain how you are assessing "tethering" and "fusion" experimentally-e.g., if fusion is defined as loss of Sec4-GFP signal at the plasma membrane, that's an important assumption that should be stated and justified.– On a related note, the manuscript would benefit from a mechanistic diagram that pulls together the new findings into a model. Some speculation is fine here. The idea is to emphasize how the results deepen our understanding of the journey experienced by a secretory vesicle.

We have added a table up front to succinctly summarize proteins discussed within the text, though we have relied on in-text citations to back up-the protein functions mentioned in the table, as citations here would be unsightly. We now define tethering as "the first frame where a Sec4 vesicle came to a full stop at a location not more than one apparent vesicle diameter from where it eventually fused with the plasma membrane" and fusion "at the last frame in which signal could be positively identified at the tethering location". Later in the results, we underscore this assignment of fusion as the length of tethering is unchanged in GDI mutants where some of the Sec4 becomes incorporated into the plasma membrane (Figure 4 D,E).

Additionally, we have added a summary figure schematizing the timeline of events as the new Figure 9 within the discussion and clarified some assumptions with regards to Figure 6.

1) Given the emphasis in this work on imaging technology, it would be useful to provide full explanations about how the microscopy system was configured and why particular choices were made.

As no new imaging technology was employed, simply optimized techniques and newer cameras, we did not believe a mechanistic diagram of imaging technology would be helpful, but we have elaborated further as to the choices made and the set-up of the system within the methods.

2) Studies of related phenomena in Aspergillus should be cited and considered. It's likely that many basic aspects of secretion are conserved between yeast and Aspergillus, but the Aspergillus system has unique properties that can illuminate features of the pathway.

We have incorporated into the discussion differences in the TGN-to-secretory vesicle transition observed in Aspergillus nidulans, while showing that this system also employs the Mso1 protein and that this protein is not missing from filamentous fungi as one reviewer suggests, but is rather isolated to ascomycetes.

3) Further discussion of Ypt31/32 would be helpful. The reviewers were surprised by the conclusion that Ypt31/32 remains on secretory vesicles. Perhaps the GFP tag partially compromises function and inhibits GAP activity? I note that GFP tags have been reported to compromise Ypt1 function, and we see something similar for Ypt6. Another point is that the Ypt31/Ypt32 pair arose from a relatively recent whole-genome duplication, so it's unlikely that the two paralogs have substantially different functions or that such differences are relevant beyond yeast. This concept is worth mentioning.

Ypt31/32 are likely not significantly enriched on the plasma membrane post vesicle fusion. The concern that the fluorescent tags used could be compromising Ypt31/32 function is a valid one and one that we are certainly sensitive to. Upon discussion with colleagues and review of early literature regarding Ypt31 localization we have decided to deemphasize this observation as at least its localization to the plasma membrane is likely an artifact. We still, however, believe that Ypt31 and Ypt32 may be playing subtly different cellular roles. Though clearly the two provide similar-enough functions to compensate for the loss of one another, of the 54 reported significant negative genetic interactors of Ypt32, only two overlap with significant negative genetic interactors of Ypt31 (of 182 total). Similarly, the two show no overlap in significant positive genetic interactions. (Costanzo M et al. Science. 2016 Sep 23;353(6306). pii: aaf1420. PubMed PMID: 27708008.)

4) It would be helpful to distinguish more clearly between firm interpretations and speculations, especially in cases where you are pushing the technology to the limit and making inferences about what you are seeing.

As indicated in the responses to reviewers, we have tried to distinguish between firm interpretations and speculation.

5) Some of the conclusions echo prior work from other investigators, and those earlier contributions should be better acknowledged. A particular example is the idea that Sec3 is not actually a landmark component (see Roumanie et al., 2005).

We apologize for this omission, and now clearly highlight this contribution.

Reviewer #1 (Recommendations for the authors):"TRAPPII" has two P's.

Thanks for spotting this error.

Regarding the subtle differences between Ypt31 and Ypt32: is it possible that Ypt32 associates with secretory vesicles more efficiently than Ypt31, with the result that the local concentration of Ypt32 in the bud is higher than that of Ypt31?

Yes, this is possible, but we believe the difference is of-note regardless of the cause.

Line 188: "Figure 5D" should presumably read "Figure 2D".

Thanks for spotting this error.

I don't quite understand Figure 2E. Isn't Sec4 present on the vesicle before tethering? If so, why is the Sec4 intensity very low just prior to tethering? Maybe the zero on the horizontal axis is in the wrong place?

The zero is indeed mis-placed, and a revised figure now replaces Figure 2E.

In Figures 4, 7, and 8, I believe the convention for yeast is to capitalize all letters of a wild-type gene name rather than just the first letter.

Thank you – this issue has been fixed in the revised figures.

Reviewer #2 (Recommendations for the authors):Additional discussions of the generality of these findings, e.g. will this information be similar for non-yeast eukaryotes, would be helpful.

We now have a new figure (Figure 9) indicating the sequence of events. As this is the first-ever time-line, little can be speculated about how it might relate to non-yeast eukaryotes.

Could the Ypt31 on the plasma membrane (or near the plasma membrane) represent recycling vesicles?

Indeed, we believe this may be the case and this is now mentioned in the Discussion.

For the Exocyst-3x-tag construct, why were the 3 subunits all chosen to be from the same subcomplex rather than some from each subcomplex? Presumably, the exocyst is octameric, but this construct would only report on half of it.

Sec10 and Sec15 were chosen first. For issues of sterics alone, we initially excluded Sec5 and Exo70 from a multi-tag complex. We also excluded Sec6 due to critical interactions with SNAREs and Sec1 which we feared interfering with (the exact residues required for these interactions are unknown). Sec3 and Sec8 may have worked fine as alternative tags and, in fact, we wanted to use a 4x-tag exocyst, but found this behaved worse than the 3x. Exo84 ended up as the 3^rd^ tag by happenstance. In retrospect at least one on the other exocyst <milestone-start />“<milestone-end />half” would have been nice, but it did not end up being that way.

The Rho3 timing vs. exocyst seems to be too noisy to be able to clearly conclude that Rho3 peaks ~1 sec after tethering starts.

Yes, this is an estimation and we now say this more clearly, but a major point is that local Rho3 levels rise and fall.

The genetic experiments (heterozygous and homozygous deletions, and overexpression) are interesting, but difficult to make unambiguous conclusions from, especially if the levels of the proteins in the heterozygous cells are unknown, and relative overexpression levels vs endogenous proteins are also unknown.

Yes, this is always a concern when analyzing heterozygous cells (although levels generally go down two-fold) and over-expression studies, where again levels generally go up substantially. We do not have the reagents to directly assess these levels.

In line 482, the reference for the "new observations" paper is missing.

This doesn't refers to a published study, but to the results that follow.

The overall writing could be tightened up, and several typos are scattered throughout.

We tried to write this in a way that is easy to follow, especially given the complexity of the system. We have double-checked for typos.

Reviewer #3 (Recommendations for the authors):– Ypt31 but not Sec7 remaining upon vesiculation? This does not match with the fact that Sec7 is a Ypt31 effector (Fromme, Dev Cell)

Sec7 is indeed an effector of Ypt31, but not in a linear timeline. Instead it interacts though a feedback mechanism. It is possible that some Sec7 remains until the moment of vesicle fission, but it does not appear to be there in abundance if this is the case.

– Figure 1B tracking of SEC2 not convincing, perhaps related to timestamp differences between dual-color images. Figure 6C tracking of Rho3 is not convincing either. Why not use a beam splitter?

As noted above, our microscope is not configured with a beam splitter.

– Please clarify further if there are separated roles for Ypt31 and Ypt32.

We have no information about whether Ypt31 and Ypt32 play different roles; we only note that their distributions are not identical, and as indicated above, discuss possible distinct functions.

– Please integrate into the timeline the fact that Myo V is a direct effector of Ypt31/32 (Lipatova and Segev, 2008) lines 99-100.

This was an unintentional omission and is now mentioned.

– Please explain why a 'sizeable population of Ypt31 remains on the membrane' (Figure 1A).

As indicated above, we consider the possibility that this might be an artifact of tagging.

– Line 128 Hugh Pelham denoted these compartments as the post-Golgi endosome, fed by membranes derived both from the Golgi and the plasma membrane. They might be equivalent to rapid recycling endosomes of metazoans.

This is indeed what we believe and is now stated more clearly in the Discussion.

– It is crucial to establish whether detachment of Sec2 compartments occurs before motor engagement, thus this part should be further expanded, or removed if inconclusive

We are not sure what concern the reviewer has or how s/he wants it expanded. We find that Sec2-mNG arrives 2s before detachment and Myo2 about 0.5s, so it seems that Myo2 arrives before Sec2 detachment.

– 211 alone alone repetition.

Thanks, the error has been corrected.

– 327 why very few? compared to what?

We will clarify this, but the 'very few' is in compariosn to the number of copies (~70) of Sec4

– 338 perturbation (typo); does it mean synthetic negative interaction? why 'apparent'?

Thanks for spotting the typo. The word apparent was not appropriate; what we meant was that tagging two different components showed deleterious growth effects, whereas single tagging did not.

– Figure 6S2, add linescan.

We are not sure why this is necessary

– how do you power vesiculation?

We don't believe there is any information about this, except to note that it happens in the asnece of actin filaments.

– 420- lack of effect of Sec2 overexpression: GFP-Sec4 insensitive to nt exchange?

We were surprised by this as well. It may play a larger role in the initial recruitment of Sec4 and not on tethering time, likely due to how quickly Sec2 dissociates from the vesicles after reaching or nearing the destination.

– Figure 7S1 B MyoE facilitates (shortens) the tether to fusion transit; how do you envisage it mechanistically?

We have added a statement regarding this.

– Please provide a readable file 8S2. The wording is confusing here, yeast vs. fungi? Mso1 appears to be absent from filamentous fungi, you may cite FungiDB (https://fungidb.org/fungidb/app/record/gene/YNR049C#Orthologs).

As noted above, we have expanded this analysis, and in fact Mso1 is in filamentous fungi.

– I missed the evidence that Mso1 is a Sec4 effector – Figure 8I?

This has been reported by Weber-Boyvat et al., 2011 which is now cited here.

– 501… large number of secretory vesicles per unit time: would authors endeavor to provide an estimation?

We have added a discussion of this.

– 530: a beam splitter may have been useful.

Yes, but we don’t have one.

– 534 Ypt31 marked secretory compartments might mislead readers.

Thanks, we have changed the wording.

– 578 Has it been formally established that Q79L prevents GTP hydrolysis? (see Barr, 2014, Diversity and plasticity in Rab GTPase nucleotide release mechanism has consequences for Rab activation and inactivation)

We do not say Sec4-Q49L has no GTP hydrolysis. We say discuss the effects of 'delayed' hydrolysis.